# STR2STR: A SCORE-BASED FRAMEWORK FOR ZERO-SHOT PROTEIN CONFORMATION SAMPLING

**Jiarui Lu[1,2], Bozitao Zhong[1,2], Zuobai Zhang[1,2], Jian Tang[1,3,4]**
[1]Mila - Québec AI Institute, [2]Université de Montréal
[3]HEC Montréal, [4]CIFAR AI Chair
{jiarui.lu, bozitao.zhong, zuobai.zhang}@mila.quebec,
jian.tang@hec.ca

## ABSTRACT

The dynamic nature of proteins is crucial for determining their biological functions and properties, for which Monte Carlo (MC) and molecular dynamics (MD) simulations stand as predominant tools to study such phenomena. By utilizing empirically derived force fields, MC or MD simulations explore the conformational space through numerically evolving the system via Markov chain or Newtonian mechanics. However, the high-energy barrier of the force fields can hamper the exploration of both methods by the rare event, resulting in inadequately sampled ensemble without exhaustive running. Existing learning-based approaches perform direct sampling yet heavily rely on target-specific simulation data for training, which suffers from high data acquisition cost and poor generalizability. Inspired by simulated annealing, we propose STR2STR, a novel structure-to-structure translation framework capable of zero-shot conformation sampling with roto-translation equivariant property. Our method leverages an amortized denoising score matching objective trained on general crystal structures and has no reliance on simulation data during both training and inference. Experimental results across several benchmarking protein systems demonstrate that STR2STR outperforms previous state-of-the-art generative structure prediction models and can be orders of magnitude faster compared to long MD simulations. Our open-source implementation is available at https://github.com/lujiarui/Str2Str.

## 1 INTRODUCTION

Understanding the dynamical properties of proteins is crucial for elucidating the mechanism of their biological functions and regulations. Transitions can exist in the conformational ensemble, ranging from angstrom to nanometer in length, and from nanosecond to second in time. Experimental measurements, such as crystallographic B-factors and NMR spectroscopy, can be used to probe such dynamics yet in limited spatial and temporal scale. Despite the success of structure prediction models (Baek et al., 2021; Jumper et al., 2021; Lin et al., 2023) which enables the study of proteins based on high-accuracy structures, the predicted ensembles often lack diversity (Chakravarty & Porter, 2022; Saldaño et al., 2022) and modeling structure-dynamics relationship remains a challenge.

Traditionally, Monte Carlo (MC) and molecular dynamics (MD) are two predominant families for conformation sampling by employing an empirical force field. Both of them operate by starting from an initial point and exploring the conformation space guided by the force field. MC methods sample conformations by steering a Markov chain of stochastic perturbations (eg., Gaussian noise) on the Cartesian or internal coordinates with an acceptance ratio, or Markov chain Monte Carlo (MCMC). However, the transition kernel can rapidly lose exploration efficiency with an increasing degree of freedom. On the other hand, MD simulations evolve the motion of atoms over time to generate time-indexed trajectories via the Newtonian mechanics. Due to the tiny timestep, a significant challenge encountered by MD simulation is the high energy-barrier, which forbids thermodynamics-favored transitions within a limited number of simulation steps. To ameliorate, enhanced sampling methods have been proposed to overcome the energy barrier and encourage more exploration of MD simulations. For example, methods based on biased potentials, such as umbrella sampling (Torrie & Valleau, 1977) and metadynamics (Laio & Parrinello, 2002); and those inspired by simulated

annealing that schedule the temperature to encourage exploration, e.g., replica exchange molecular dynamics or REMD (Hansmann, 1997; Sugita & Okamoto, 1999; Swendsen & Wang, 1986).

Another increasingly appealing solution to the problem is the generative modeling of protein conformations. Direct sampling by the neural generator is more efficient than time-consuming simulations from MC or MD. Boltzmann generator (Noé et al., 2019), as one of the earliest attempt, modelled the system-specific conformation distribution with normalizing flow and performed *i.i.d.* sampling from random noises. With reweighting, the sampled ensemble can approximate the physical Boltzmann distribution. However, learning from a specific protein system requires pre-acquired simulation data for training the sampler and can be difficult to generalize beyond the training system (Wang et al., 2019), leaving the use of such methods limited. Although generative training on the across-system conformation datasets can help, the data acquisition can be non-trivial due to lack of open-source MD trajectories for protein systems and the computationally intensive simulations from scratch.

To address the aforementioned issues, we propose a new framework that samples general protein conformations via an equivariant structure-to-structure (STR2STR) translation. Trained on general crystal structures, STR2STR has no reliance on the computationally intensive simulation data and thus performs zero-shot[1] conformation sampling for any unseen protein. Specifically, we formulate the conformation sampling task as a translation problem within the conformation space of the target protein. Motivated by simulated annealing, the proposed translation is composed of stochastic perturbations followed by the score-based annealing, forming a forward-backward process. As an illustration, we present the inference diagram of STR2STR in comparison with three traditional methods in Figure 1. We demonstrate that the sampling process is equivariant to global roto-translations of the protein geometry, which guarantees the inference not yielding samples as trivial as rotated or translated variants. For evaluation, we construct a benchmark covering various aspects for protein conformation sampling and perform a case study of protein BPTI to demonstrate the effectiveness of STR2STR. Experimental results show that our method not only significantly outperforms the previous baselines on protein conformation sampling but is also comparable to long MD simulations.

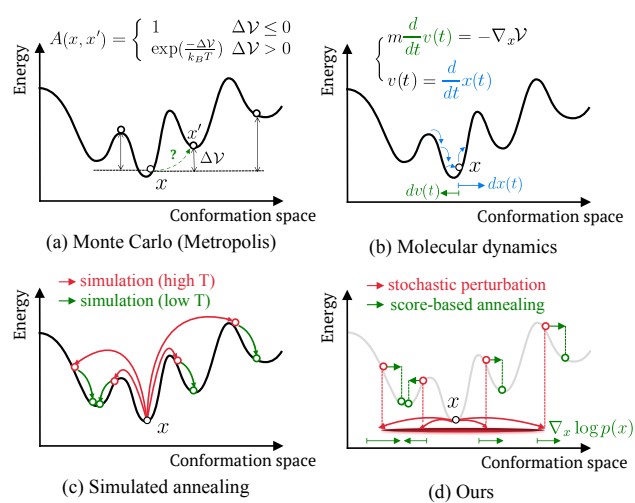

Figure 1: Illustration of traditional sampling methods with proposed STR2STR. In (d), the energy landscape is made transparent to indicate that, in contrast to traditional cases, STR2STR is agnostic to and thus not relying on the energy landscape but guided by the learned score functions.

## 2 PRELIMINARIES

**Equivariance of transformation.** Equivariance of a function (mapping) indicates that applying specific transformations (for example, rotation for Euclidean space) to the input or output of a function should have corresponding effects on the final output value. Formally, a function $\mathcal{F} : \mathcal{X} \to \mathcal{Y}$ with equivariant property can be described as:

$$\mathcal{F} \circ \rho(x) = \rho \circ \mathcal{F}(x), \tag{1}$$

where $\rho$ is some transformation which acts on the element from space $\mathcal{X}$ or $\mathcal{Y}$.

---

[1]In the context of this paper, **zero-shot** means having no access to simulation data that belongs to the test protein during both training and inference stage.

**Diffusion modeling on Riemannian manifolds.** Score-based generative models (SGMs) can be represented by a diffusion process $\mathbf{x}_t \in \mathbb{R}^n$ defined by the Itô stochastic differential equation (SDE):

$$\mathrm{d}\mathbf{x} = \mathbf{f}(\mathbf{x}, t)\mathrm{d}t + g(t)\mathrm{d}\mathbf{w}, \tag{2}$$

with continuous time index $t \in [0, T]$, where the $\mathbf{f}(\mathbf{x}, t) \in \mathbb{R}^n$ is the drift term, $g(t) \in \mathbb{R}$ is the diffusion coefficient, and $\mathbf{w} \in \mathbb{R}^n$ is the standard Wiener process (or Brownian motion). Then, the corresponding backward SDE that describes the dynamics from $\mathbf{x}_t$ to $\mathbf{x}_0$ is (Anderson, 1982; Song et al., 2020):

$$\mathrm{d}\mathbf{x} = [\mathbf{f}(\mathbf{x}, t) - g^2(t)\nabla_{\mathbf{x}} \log p_t(\mathbf{x})]\mathrm{d}t + g(t)\mathrm{d}\bar{\mathbf{w}}, \tag{3}$$

where $\mathrm{d}t$ is negative infinitesimal timestep and $\bar{\mathbf{w}}$ is the standard Wiener process as continuous time $t$ flows back from $T$ to $0$. De Bortoli et al. (2022) proposed the corresponding forward and backward process on a Riemannian manifold $\mathcal{M}$ beyond Euclidean space. To steer diffusion process with validity, the drift $\mathbf{f}(\mathbf{x}, t)$, Brownian motion $\mathbf{w}$, and score $\nabla_{\mathbf{x}} \log p_t(\mathbf{x})$ are elements in the tangent space $T_{\mathbf{x}}\mathcal{M}$. Utilizing the exponential-logarithm map, the process can be discretized similar to the Euler–Maruyama step in Euclidean space as geodesic random walk. Several recent works realized the Riemannian diffusion for different types of geometric data. Jing et al. (2022) constructed the torsional diffusion on a hypertorus $\mathbb{T}^n$ while Yim et al. (2023) developed the $\mathrm{SE}(3)^n$ diffusion for orientation-preserving rigid motions in 3D space [2].

**Notation on protein structure.** The protein conformation is represented by its Euclidean coordinates $\mathbf{x} \in \mathbb{R}^{3 \times N}$, where $N$ is the number of heavy atoms (excluding hydrogen). We adopt the *backbone frame* parametrization $\mathrm{T}_i := [R_i, \boldsymbol{v}_i](1 \leq i \leq n)$ one per residue . Here, $R_i \in \mathrm{SO}(3)$ is a $3 \times 3$ rotation matrix while $\boldsymbol{v}_i \in \mathbb{R}^3$ is a translation vector for the $i$-th residue. Such tuple represents an Euclidean transformation for each atom $\boldsymbol{x}$ in residue $i$ from the local coordinate $\boldsymbol{x}_{\mathrm{local}} \in \mathbb{R}^3$ to a position in global coordinates as $\boldsymbol{x}_{\mathrm{global}} = \mathrm{T}_i \circ \boldsymbol{x}_{\mathrm{local}} := R_i \boldsymbol{x}_{\mathrm{local}} + \boldsymbol{v}_i$. The global atom coordinates on the backbone, specifically $[\mathrm{N}, \mathrm{C}^{\alpha}, \mathrm{C}, \mathrm{C}^{\beta}]$ (except for GLY which has no $\mathrm{C}^{\beta}$), can be constructed by applying the transformation induced by $\mathrm{T}_i$ to the corresponding amino acid structure with idealized bond length and angles (Jumper et al., 2021), that is $\mathbf{x}_{bb} = \Gamma_{bb}(\{\mathrm{T}_i\})$, where $\{[\cdot]_i\} := ([\cdot]_1, \ldots, [\cdot]_n)$ is a brief sequence notation and $\Gamma_{bb}(\cdot)$ constructs the corresponding global coordinates. Conditioned on $\mathbf{x}_{bb}$, the carbonyl oxygen on backbone can be parameterized by a torsion angle $\psi_i$, or written as $\mathbf{x}_{bb[\mathrm{O}]} = \Gamma_{bb[\mathrm{O}]}(\{\psi_i\}; \mathbf{x}_{bb})$. The side chain coordinates of $i$-th residue can be parameterized by at most four torsion angles $\boldsymbol{\chi}_i := (\chi_1, \chi_2, \chi_3, \chi_4)_i \in [0, 2\pi)^4$, according to the rigid groups on which these heavy atoms depend. For example, in amino acid proline (PRO), the $\mathrm{C}^{\delta}$ atom belongs to its $\chi_2$-group, which further depends on $\chi_1$ and $\chi_2$ (see Appendix A.2 for the full rigid group definition). Given the backbone coordinates, the Euclidean coordinates of side chains can be constructed with these torsion angles, which is denoted as $\mathbf{x}_{sc} = \Gamma_{sc}(\{\boldsymbol{\chi}_i\}; \mathbf{x}_{bb})$. Finally, we write collectively $\mathbf{T} := \{\mathrm{T}_i\}$, $\mathbf{R} := \{R_i\}$, $\mathbf{v} := \{\boldsymbol{v}_i\}$, $\boldsymbol{\psi} := \{\psi_i\}$ and $\mathcal{X} := \{\boldsymbol{\chi}_i\}$.

## 3 METHODS

Conformation sampling involves learning the probability distribution $p_X(\mathbf{x})$ of some protein $X$ and then drawing samples $\mathbf{x} \sim p_X(\mathbf{x})$. Different from organic molecules whose stable conformers are relatively more constrained (Jing et al., 2022), the conformation data of protein is however intractable to acquire due to the complexity of protein systems. Secondly, modeling protein directly in atomic level can be difficult due to the scaling of the number of atoms: protein with merely 60 residues can contain roughly ∼500 heavy atoms without considering hydrogens. To address the challenges above, we propose to approach the conformation sampling by transfer learning via a translation proposal on the residue frames, which is detailed as follows: Section 3.1 formulates the modeling of probability distribution; Section 3.2 introduces the sampling framework and model architecture; Section 3.3 describes the amortized learning objectives.

### 3.1 CHAIN RULE OF THE TRANSLATION DISTRIBUTION

Given an initial conformation, the goal of conformation sampling is to capture the underlying dynamics of the target protein and infer plausibly stable candidates. We represent the overall

---

[2]Note that the frame $\mathrm{T}_i \in \mathrm{SE}(3)$ is the data point. Some literature mentioned "SE(3)-equivariance" as the function equivariance to all the (global) rotations and translations in 3D space. To avoid ambiguity, we refer to the latter as *roto-translation equivariance*.

translation distribution as $p_X(\mathbf{x}|\mathbf{x}_0)$, with $\mathbf{x}_0$ being an initial structure of protein $X$. Due to the enormous degrees of freedom in the atomic structure, direct modeling and sampling from $p_X(\mathbf{x}|\mathbf{x}_0)$ can be intractable. Based on the structural hierarchy, we decompose $p_X(\mathbf{x}|\mathbf{x}_0) = p_X(\mathbf{x}_{sc}|\mathbf{x}_{bb},\mathbf{x}_0)\,p_X(\mathbf{x}_{bb[\mathrm{O}]}|\mathbf{x}_{bb},\mathbf{x}_0)p_X(\mathbf{x}_{bb}|\mathbf{x}_0)$, The rationale behind is that: given the backbone, the corresponding side chains take relatively limited orientations and can be sampled more efficiently. Therefore, the sampling can be performed step-wise: firstly, backbone frames are sampled from the backbone proposal $\mathbf{T} \sim p_X(\mathbf{T}|\mathbf{x}_0)$, and backbone coordinates can be obtained followed by the local-to-global construction $\mathbf{x}_{bb} = \Gamma_{bb}(\mathbf{T})$. Secondly, the torsion angles can be sampled conditioning on the coordinates of backbone atoms: $\boldsymbol{\psi} \sim p_X(\boldsymbol{\psi}|\mathbf{x}_{bb},\mathbf{x}_0)$ and $\mathcal{X} \sim p_X(\mathcal{X}|\mathbf{x}_{bb},\mathbf{x}_0)$. Since the torsion angles are usually treated as internal coordinates, we may assume that these conditional torsion proposals only depend on the sampled backbone itself, i.e, $p_X(\boldsymbol{\psi}|\mathbf{x}_{bb},\mathbf{x}_0) \approx p_X(\boldsymbol{\psi}|\mathbf{x}_{bb})$ and $p_X(\mathcal{X}|\mathbf{x}_{bb},\mathbf{x}_0) \approx p_X(\mathcal{X}|\mathbf{x}_{bb})$, the backbone oxygen and side chain atoms can be constructed as follows $\mathbf{x}_{bb[\mathrm{O}]} = \Gamma_{bb[\mathrm{O}]}(\boldsymbol{\psi},\mathbf{x}_{bb}), \mathbf{x}_{sc} = \Gamma_{sc}(\mathcal{X},\mathbf{x}_{bb})$ and finally $\mathbf{x} = [\mathbf{x}_{bb},\mathbf{x}_{bb[\mathrm{O}]},\mathbf{x}_{sc}] \sim p_X(\mathbf{x}|\mathbf{x}_0)$.

## 3.2 EQUIVARIANT STRUCTURE-TO-STRUCTURE TRANSLATION

**Forward-backward Process.** To model the backbone proposal distribution, we firstly consider the distribution over Riemannian manifold $\mathrm{SE}(3)^n$ where length-$n$ frame sequences $\mathbf{T}$ populates. We firstly make a mild assumption that the proposal can be approximated by removing the initial side chain dependency $p_X(\mathbf{T}|\mathbf{x}_0) = p_X(\mathbf{T}|\mathbf{T}_0,\psi_0,\mathcal{X}_0) \approx p_X(\mathbf{T}|\mathbf{T}_0)$, which forms a translation problem[3] within the space of $\mathrm{SE}(3)^n$. Motivated by simulated annealing, we propose a general score-based forward-backward (FB) process[4] that mimics the heating and annealing process. Here, the perturbing (heating) process aims to enhance the exploration while the annealing guarantees the fidelity (fine-grained

Figure 2: Illustration of forward-backward process. Given an input structure (example as Trp-cage, PDB entry: 2JOF), replicas are fed to the forward (*perturb*) diffusion, which independently perturbs each replica until the dynamic-transition time $T_\delta$; then the reverse (*anneal*) process will yield the sampled conformations. The sequence-to-structure task can be well solved by any existing folding module such as ESMFold.

structural characteristics) by exploitation. In practice, the FB process leverages a stochastic perturbation kernel and multi-scale score functions, or formally defined by the following integrals:

$$\mathbf{T} := \mathbf{T}_0 + \int_0^{T_\delta} [\mathbf{f}(\mathbf{T}_t,t)\mathrm{d}t + g(t)\mathrm{d}\mathbf{w}]$$
$$+ \int_{T_\delta}^{2T_\delta} \left\{ \left[-\mathbf{f}(\mathbf{T}_\tau,\tau) + g^2(\tau)\nabla_{\mathbf{T}_\tau}\log p_\tau(\mathbf{T}_\tau)\right]\mathrm{d}\tau + g(\tau)\mathrm{d}\bar{\mathbf{w}} \right\}, \tag{4}$$

where $\tau = \tau(t) := 2T_\delta - t\ (T_\delta \in (0,T))$ is the change of time variable and the rest of symbols are defined similarly in Eq. (2) and (3). Here the addition operator indicates the composition of frames and updates symbolically. Intuitively, the Eq. (4) perform noise injection (forward) followed by denoising process (backward) belonging to the above diffusion process defined on the manifold of $\mathbf{T}$. The bound of integration $T_\delta$ is set to be strictly less than $T$ limiting the perturbation scale not to eliminate the information of the initial condition $\mathbf{T}_0$. Empirically, increasing $T_\delta$ to a proper extent can lead to enhanced diversity yet it may hurt exploitation by demanding more reverse steps.

**Diffusion Process on $\mathrm{SE}(3)^n$.** The diffusion process $(\mathbf{T}_t)_{t\in[0,T]} \equiv ([\mathbf{R}_t,\mathbf{v}_t])_{t\in[0,T]}$ defined on manifold $\mathrm{SE}(3)^n$ can be represented as follows, by treating $\mathrm{SO}(3)$ and $\mathbb{R}^3$ independently (Yim et al.,

---

[3]In analogy to text-to-text translation and image-to-image translation.

[4]Experiments in this work only involve sampling from an identical input structure. However, it is natural to enforce FB sequentially as a neural proposal in MCMC. We leave this for future work.

2023):

$$d\mathbf{T}_t = [0, -\frac{1}{2}\beta(t)\boldsymbol{P}\mathbf{v}_t]dt + [\sqrt{\frac{d}{dt}\sigma^2(t)}d\mathbf{w}^{(\mathrm{SO}(3))}, \sqrt{\beta(t)}\boldsymbol{P}d\mathbf{w}^{(\mathbb{R}^3)}], \tag{5}$$

where $\beta(t), \sigma(t) \in \mathbb{R}_+$ are diffusion noise schedules, $\mathbf{w}^{(\mathcal{M})}$ indicates the Brownian motion defined on manifold $\mathcal{M}$ and the projection matrix $\boldsymbol{P} : \mathbb{R}^{3n} \to \mathbb{R}^{3n}$ removes the center of mass. The perturbation kernel $p_{t|0}(\mathbf{R}_t|\mathbf{R}_0)$ for the rotation components $(\mathbf{R}_t)_{t\in[0,T]}$ is considered element-wise via the isotropic Gaussian on SO(3) distribution (Leach et al., 2022; Yim et al., 2023):

$$\mathcal{IG}_{\mathrm{SO}(3)}(R_t; R_0, \sigma^2) := f(\omega_{t|0}) := \frac{1-\cos(\omega_{t|0})}{\pi} \sum_{l=0}^{\infty} (2l+1) e^{-l(l+1)\sigma^2} \frac{\sin((l+\frac{1}{2})\omega_{t|0})}{\sin(\frac{\omega_{t|0}}{2})}, \tag{6}$$

with $\omega_{t|0} = \mathrm{Axis\_angle}(R_0^\top R_t)$ is the axis-angle representation of the composed rotation matrix $R_t^\top R_0$. On the other hand, the perturbation kernel for translation components $(\mathbf{v}_t)_{t\in[0,T]}$ is an Ornstein-Uhlenbeck process, also known as VP-SDE (Song et al., 2020), which induces the isotropic gaussian kernel $p_{t|0}(\mathbf{v}_t|\mathbf{v}_0) = \mathcal{N}(\mathbf{v}_t; \mathbf{v}_0 e^{-\frac{1}{2}\int_0^t \beta(s)ds}, \mathbf{I} - \mathbf{I}e^{-\int_0^t \beta(s)ds})$ and converges to $\mathcal{N}(\mathbf{0}, \mathbf{I})$.

**Packing of Side Chains.** Given the sampled frames, we can construct the atom coordinates on the backbone and then sample the side chains from $p_X(\mathcal{X}|\mathbf{x}_{bb})$. Traditionally, this has been formulated as the protein side chain packing (PSCP) task (Xu & Berger, 2006). PSCP aims to, instead of freely exploring the conformation space, finding the conformation of side chains that minimize the energy. This casts the generative modeling of $p_X(\mathcal{X}|\mathbf{x}_{bb})$ into its discriminative form, i.e. regression of torsion angles. In practice, we adopted the FASPR (Huang et al., 2020), an efficient open-source method that leverages the backbone-dependent rotamer libraries and a simulated annealing Monte Carlo searching scheme to predict the most probable side chain conformations.

**Roto-translations Equivariance.** Consider the forward-backward process in Eq. (4). The $\mathrm{SE}(3)^n$ diffusion integral in Eq. (5) only updates the local-to-global transformations induced by the frames $\mathbf{T}$, and therefore the equivariance holds due to that fact that both drift and diffusion terms in Eq. (5) are frame-independent. For the backward integral, the extra term in the integral is the frame-dependent score function $\nabla_{\mathbf{T}_t} \log p_t(\mathbf{T}_t)$. Based on the result above, if the score function is equivariant, the reverse diffusion as well as the whole forward-backward process are equivariant. The equivariance of packing steps naturally holds because the predicted torsion angles are naturally internal coordinates and roto-translation invariant. Therefore, we can derive the following proposition:

**Proposition 1** (Equivariance of STR2STR). *Let* $\mathbf{x} \sim p_X(\mathbf{x}|\mathbf{x}_0)$ *be the conformation sampled from the process defined in Section 3.1. If the frame score functions* $\nabla_{\mathbf{T}_t} \log p_t(\mathbf{T}_t)$ *are equivariant to global roto-translations, then* $\mathbf{x}_0 \to \mathbf{x}$ *assumes roto-translation equivariance.*

The detailed proof of Proposition 1 can be found in Appendix B.

**Score Network Architecture.** To model the translation distributions, the score model is required to obey the equivariant property with respect to global rotations and translations. We adopted a variant of the structure module in Jumper et al. (2021) called DenoisingIPA, to predict the score and steer the backward diffusion process. In DenoisingIPA, we initialize the single embedding $\{\boldsymbol{s}_i\}^0$ as the concatenation of the position encoding of residues and sinusoidal time embedding; the pair embedding $\{\boldsymbol{z}_{ij}\}^0$ is constructed from the relative positional encoding (Shaw et al., 2018). In each layer $l$, the single representation $\{\boldsymbol{s}_i\}^l$ and frames $\{\mathrm{T}_i\}^l$ are updated via the Invariant Point Attention (IPA) layer and backbone update (Algo. 22-23 in Jumper et al. (2021)), followed by the multi-head self-attention (Vaswani et al., 2017) and multiple layer perceptrons (MLP). The update of representations and frames are illustrated in Figure 3. Slightly different from the vanilla structure module, we also allow the update of pair representations $\{\boldsymbol{z}_{ij}\}^l$ by edge transition layers:

$$\{\boldsymbol{z}_{ij}\}^{l+1} = \mathrm{MLP}\left(\mathrm{Concat}\left[\{\boldsymbol{z}_{ij}\}^l, \{\boldsymbol{s}_i^l \otimes \boldsymbol{s}_i^l\}\right]\right), \tag{7}$$

where $\otimes$ indicates the outer product. Following Jumper et al. (2021), we leverage the single representations $\{\boldsymbol{s}_i\}^L$ from the last layer to predict angle $\psi$ with an MLP. Because the carbonyl oxygen atoms do not affect global geometry, we treat it in a discriminative manner similar to side chains.

### 3.3 AMORTIZED LEARNING OBJECTIVES

**Amortized Score Matching Loss.** To learn the translation distribution $p_X(\mathbf{T}|\mathbf{T}_0)$ over the manifold $\mathrm{SE}(3)^n$, the conformation samples of $X$ are required for training the score networks. However,

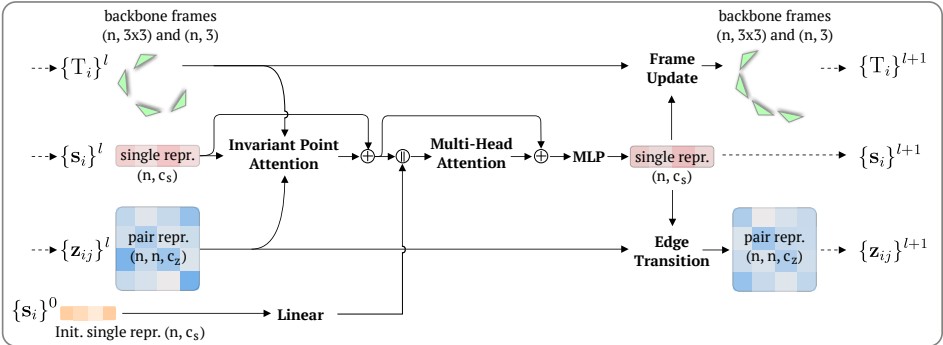

Figure 3: Illustration of $l$-th layer of DenoisingIPA, where $\|$ denotes the tensor `Concat` and $+$ means tensor `Add` operation. The multi-head attention is the transformer self-attention (Vaswani et al., 2017). The initial single representations $\{s_i\}^0$ are constructed from the positional encoding of residues and the time encoding of denoising step. Here, single representations $\{s_i\}^l$ and backbone frames $\{T_i\}^l$ are updated similar to the structure module in Jumper et al. (2021), while pair representations $\{z_{ij}\}^l$ are updated according to Eq. (7).

acquiring simulation training set suffers from high computation cost and the resulting generalization capacity is limited. To tackle this challenge, we propose to use the general crystal structures from Protein Data Bank (PDB) for training, which can be viewed as respective local minima in the energy landscape. In this amortized sense, it suffices to train a single score network for the inference of any unseen protein at test. The difference in the denoising score matching objective is that the data sample $\mathbf{T}_0$ are from general distribution of PDB (denoted as $p*$) instead of a target-specific $p_X(\mathbf{T})$:

$$\mathcal{L}_{\text{dsm}} = \mathbb{E}_{t \in [0, \tau_m]} \left\{ \lambda(t) \, \mathbb{E}_{\mathbf{T}_0 \sim p*} \mathbb{E}_{\mathbf{T}_t | \mathbf{T}_0} \left[ \| \mathbf{s}_\theta(\mathbf{T}_t, t) - \nabla_{\mathbf{T}_t} \log p_{t|0}(\mathbf{T}_t | \mathbf{T}_0) \|^2 \right] \right\}, \qquad (8)$$

where $\lambda(t) \propto 1 / \mathbb{E} \left[ \| \nabla_{\mathbf{T}_t} \log p_{t|0}(\mathbf{T}_t | \mathbf{T}_0) \| \right]$ is a positive loss reweighting function, and $\mathbf{T}_t \sim p_{t|0}(\mathbf{T}_t | \mathbf{T}_0)$ is defined by the corresponding perturbation kernel. Since the inference procedure does not require reversing from the pure random noise (when $t = T$), the time $t$ can be uniformly sampled over the truncated time domain $[0, \tau_m]$, where $0 < \tau_m \leq T$ is a pre-specified hyperparameter indicates the maximal time scale used for inference.

**Auxiliary Structural Losses** According to the findings in Yim et al. (2023), solely training by score matching can be insufficient for learning fine-grained structural characteristics. Along with the score matching loss for the frames, we complement auxiliary structural losses including mean square error (MSE) of backbone atoms and the distogram loss as in Jumper et al. (2021). MSE loss is computed over the backbone atoms (including the carbonyl oxygen) to provide supervision for prediction of $\psi$. Because the process is roto-translation equivariant and the distogram is based on the distances which are roto-translation invariant, the structural alignment is not necessary to perform. The overall training loss can be the weighted sum of all losses: $\mathcal{L} = \mathcal{L}_{\text{dsm}} + \alpha \mathcal{L}_{\text{backb}} + \beta \mathcal{L}_{\text{dist}} \ (\alpha, \beta > 0)$. The detailed definition of auxiliary losses can be found in Appendix D.

## 4 EXPERIMENTS

We compare the proposed method STR2STR to several recent baselines: MSA subsampling (Del Alamo et al., 2022), EigenFold (Jing et al., 2023), and idpGAN (Janson et al., 2023). These baselines leverage general structure datasets for training and are claimed to be able to generalize to unseen protein, which is proper for zero-shot inference. MSA subsampling (Del Alamo et al., 2022) is a AF2-based protocol to sample structure ensemble from sequence via a reduced number of recycle and subsampled multiple sequence alignments (MSA); EigenFold is a sequence-to-ensemble diffusion model trained on PDB for conditional generation of protein structures based on the sequence embeddings from OmegaFold (Wu et al., 2022b); idpGAN is a generative adversarial network (GAN) that generates sequence-conditioned conformation ensembles. For sampling of STR2STR, the initial conformation for each test target is set to be the output of ESMFold (Lin et al., 2023). Other implementation details can be found in Appendix D.

Table 1: Benchmark results of conformation sampling methods on fast folding proteins (Lindorff-Larsen et al., 2011) with reference MD trajectories. Metrics are averaged across all protein targets for each method. Reference MD data is colored brown. The ensemble from other baselines are obtained by running their codes in the standard settings. Among these metrics, Val-Clash, Val-Bond (*validity*) are the higher the better ($\uparrow$); while JS-PwD, JS-TIC, JS-Rg (*fidelity*) and MAE-TM, MAE-RMSD (*diversity*) are the lower the better ($\downarrow$). The best result from generative models is **bolded**. The JS and MAE are compared with full MD trajectories, whose blocks are thus colored grey.

| Methods | Val-Clash($\uparrow$) | Val-Bond($\uparrow$) | JS-PwD($\downarrow$) | JS-TIC($\downarrow$) | JS-Rg ($\downarrow$) | MAE-TM($\downarrow$) | MAE-RMSD($\downarrow$) |
|---|---|---|---|---|---|---|---|
| MSA subsampling | **0.999** | **0.997** | 0.634 | 0.624 | 0.656 | 0.596 | 0.713 |
| EigenFold | 0.812 | 0.874 | 0.530 | 0.497 | 0.666 | 0.448 | 0.607 |
| idpGAN | 0.960 | 0.032 | 0.480 | 0.517 | 0.661 | 0.189 | 0.592 |
| STR2STR(PF) | 0.963 | 0.992 | 0.375 | **0.397** | 0.448 | 0.150 | 0.209 |
| STR2STR(SDE) | 0.977 | 0.982 | **0.348** | 0.400 | **0.365** | **0.133** | **0.184** |
| Reference 100ns | 1.000 | 1.000 | 0.458 | 0.491 | 0.445 | 0.227 | 0.379 |
| Reference 1us | 1.000 | 1.000 | 0.317 | 0.394 | 0.303 | 0.206 | 0.339 |
| Reference 10us | 1.000 | 1.000 | 0.236 | 0.331 | 0.227 | 0.144 | 0.243 |
| Reference 100us | 0.997 | 1.000 | 0.130 | 0.155 | 0.126 | 0.063 | 0.102 |
| Reference Full | 0.997 | 1.000 | 0.000 | 0.000 | 0.000 | 0.000 | 0.000 |

## 4.1 EVALUATION METRICS

To assess the performance of STR2STR on the zero-shot conformational sampling, we set up a benchmark based on commonly used metrics in structure design and protein dynamics research. The evaluation metrics are categorized into: (a) **Validity** assesses whether the sampled conformations obey basic physical constraints; (b) **Fidelity** reflects the distributional gap between sampled ensemble and reference MD simulation (which is seen as the "ground truth"); (c) **Diversity** evaluates the possible variety of the sampled ensemble. As for reference, we set up and also benchmarked a ladder of timescales for better comparison: *100ns, 1us, 10us, 100us, full* (the longest simulation time of each target). These metrics are briefly defined as below and detailed in Appendix E:

**Validity.** The validity is defined by the ratio of conformations passing the sanity check, which examines whether the sample contains any (1) *steric clash* or (2) *broken bond*. Given a conformation sample, the steric clashes are counted by checking whether the distance of each pair of $C^\alpha$ atoms is within certain threshold that is based on atomic *van der waals* radius; while the $C^\alpha$-$C^\alpha$ "bond" is considered breaking if the distance of adjacent $C^\alpha$ atoms exceed certain threshold.

**Fidelity.** The fidelity compares the distributional divergence between the sampled ensemble and trajecotory from reference MD simulations. We adopt the symmetric Jensen-Shannon (JS) divergence based on three important quantities defined for conformations: (i) pairwise distance distribution (JS-PwD), (ii) the slowest two components of the time-lagged independent component analysis, or TICA (Naritomi & Fuchigami, 2011; Pérez-Hernández et al., 2013) (JS-TIC) and (iii) radius of gyration distribution (JS-Rg) as in idpGAN (Janson et al., 2023).

**Diversity.** The diversity can be indicated by the averaged pairwise dissimilarity scores, based on root mean square deviation (RMSD, unit: nm) and TM-score (Zhang & Skolnick, 2004). For the TM-score, we apply the inverse (i.e., $1 - \text{TM}(\mathbf{x}_i, \mathbf{x}_j)$) to express "diversity" aligned with RMSD (the higher the more diverse). We notice that the ensemble diversity is not the higher the better and depends on the characteristics of the target system. Therefore, we report the diversity difference as the mean absolute error (MAE) compared with the reference full MD simulations on both metrics.

## 4.2 FAST FOLDING PROTEINS

The benchmark set consists of 12 fast-folding protein targets with up to 1ms scale all-atom MD simulation trajectories as reference from Lindorff-Larsen et al. (2011). To evaluate on the metrics above, we generated 1,000 conformations for each target using STR2STR and other baseline models. Two different integration schemes: probability flow ("PF") and SDE are used for STR2STR. For each method, metrics are evaluated independently for each target and averaged across these targets.

The benchmarking results are shown in Table 1, from which STR2STR outperforms other zero-shot sampling baselines by a large margin. Note that EigenFold, also as a diffusion model trained on

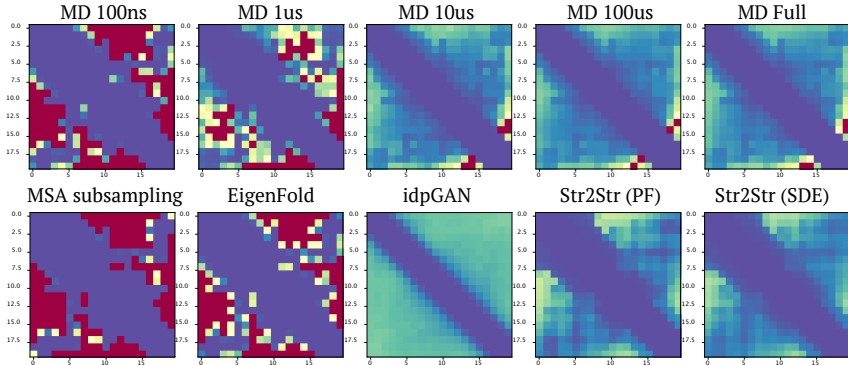

Figure 4: Contact map of Trp-cage (visualized in Figure 2) of each model with MD reference.

PDB, exhibited less diversity and failed to capture the conformational dynamics when compared with STR2STR. This may be caused by the complexity of modeling distributional mapping from sequence embedding to structure: the sequence-structure relationship can be well solved by folding models (Jumper et al., 2021; Lin et al., 2023) in a discriminative manner (regression), but is still challenging for conditional generative modeling. In contrast, the proposed sampling framework involves learning the structure-to-structure within the conformation space and learns abundant distributional features solely from the PDB database. Here we showcase the contact map of Trp-cage in Figure 4 and that of all targets can be found in Appendix F.1. The sampling speed of STR2STR with MD simulation on single GPU is shown in Table 2, where STR2STR exhibits significantly advantageous efficiency over MD simulations for a case with comparable performance. Note that in general STR2STR can still underperform long MD simulations (e.g., 100us) on the distributional metrics.

### 4.3 STRUCTURAL DYNAMICS OF BPTI

We conducted a case study using the protein Bovine Pancreatic Trypsin Inhibitor (BPTI). The dynamic characteristics of BPTI have been well studied with 1.01ms-long MD simulation in Shaw et al. (2010), based on which five kinetic clusters have been revealed. To better demonstrate the performance of STR2STR, we present the TICA plots (Pérez-Hernández et al., 2013) for the sampled conformations from each method. Specifically, the conformation coordinates are reduced to the first two TICA dimensions, which indicates the slowest two components and can embody the meta-stable states with distinction. The TICA parameters are fit

Table 2: Profiling of sampling speed between explicit solvent MD simulation for 100us and STR2STR(PF) with comparable fidelity metrics on the example target *WW domain*.

|  | MD 100us | STR2STR |
|---|---|---|
| JS-PwD (↓) | 0.399 | 0.379 |
| JS-TIC (↓) | 0.438 | 0.458 |
| JS-Rg (↓) | 0.406 | 0.402 |
| Time | >160 GPU days | 510 GPU secs |

using the the reference full MD trajectories. As shown in Figure 5, where the kinetic clusters are colored red, STR2STR successfully captured four clusters similar to 100us simulations with small variation and outperform the rest of baselines.

## 5 RELATED WORK

**Protein Backbone Design.** A parallel research interest emerging recently focuses on the protein backbone structure design based on deep generative models. Early attempts include ProtDiff (Trippe et al., 2022), which generates novel $C^\alpha$-only backbones; protein structure-sequence co-generation based on structural constraints (Shi et al., 2022); and diffusion models tailored for antibody design (Luo et al., 2022). FoldingDiff complements these by applying diffusion to the dihedral angles of backbones. Chroma (Ingraham et al., 2022) designs novel protein backbones with several conditional inputs including natural language and comprehensively evaluates the programmability. Meanwhile, RFDiffusion (Watson et al., 2022) pushed the diffusion-based protein design to the ex-

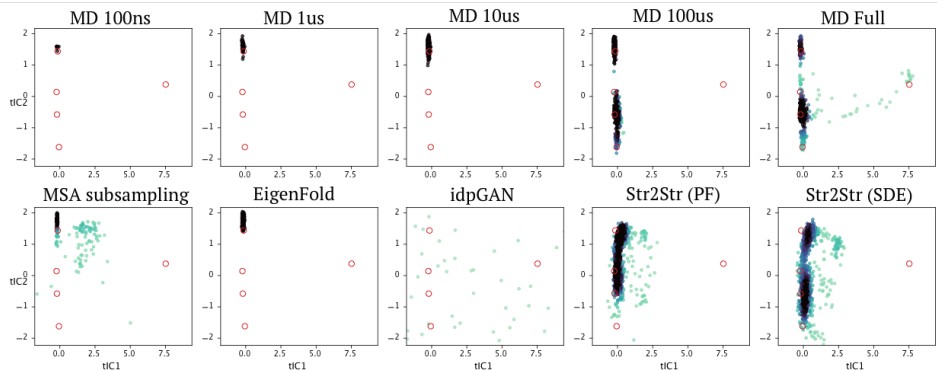

Figure 5: Visualization of TICA plots for BPTI conformations sampled by each model with MD references. The kinetic clusters are colored red. In each subfigure, totally 1,000 samples were scattered in the 2D space. Note that most of the points are outside the target region for idpGAN.

perimental side and validated the effectiveness of generative modeling for this task. More advanced methods including Genie (Lin & AlQuraishi, 2023) and FrameDiff (Yim et al., 2023) have been proposed very recently that leverages the invariant point attention modules to enhance model capacity.

**Learning from Simulation Data.** Due to the inefficiency of classical simulations for protein dynamics, several works attempted to perform efficient sampling or learn neural force fields from protein-specific simulation data. Boltzmann generators (Noé et al., 2019) were developed to generate equilibrium samples using normalizing flows (Dinh et al., 2014; Rezende & Mohamed, 2015) trained on simulation data or energy. CGNets (Wang et al., 2019) proposed to learn coarse-grained (CG) force fields in a supervised learning manner. Köhler et al. (2023) improved this by complementing density estimation and sampling right before force-matching, thus not relying on ground-truth forces in simulation data. Arts et al. (2023) proposed to train diffusion model on conformations from equilibrium distribution of a specific protein, and leveraged learned score functions as force field for simulation or i.i.d. sampler. Wang et al. (2022) attempted to recover the REMD ensembles of a small peptide by training denoising diffusion on trajectories. However, these models suffer from the transferability problem (Wang et al., 2019) and cannot be generalized to unseen proteins. Klein et al. (2023) improves by modeling transition of a large timestep in MD simulation using normalizing flow, which achieves good performance, yet only for very small peptides (only 2-4 AA). Our STR2STR is distinguished from these methods by performing zero-shot conformation sampling for unseen protein without any simulation data, and has more promising use in practice.

## 6 CONCLUSION

In this paper, we presented STR2STR, a score-based structure-to-structure translation framework for zero-shot protein conformation sampling. Motivated by simulated annealing, STR2STR tactfully combines both exploration and exploitation into a forward-backward process based on the denoising diffusion for protein frames. STR2STR was trained solely on crystal structures from the Protein Data Bank (PDB) and has no dependency on any simulation data during training or inference. Experimental results on several MD benchmarking systems demonstrate that STR2STR can effectively sample a diverse ensemble from the input structure in a zero-shot manner. Limitations and potential future directions of STR2STR encompass: **(1)** The isotropic perturbation kernels could be biased towards more efficient subspace based on some collective variables. **(2)** Since STR2STR samples all-atom conformation, it can be plugged into atom-level MD simulations by incorporating physical-based force fields and perform enhanced sampling. **(3)** The pre-trained STR2STR can be further fine-tuned by simulation data from specific systems to improve the sampling quality in a *few-shot* manner or towards the unbiased sampling from Boltzmann distribution.

REPRODUCIBILITY STATEMENT

For reproducibility, we provide the detailed implementation details and training procedures in Appendix D. To describe the proposed forward-backward sampling process, a pseudo-code snippet is shown in Algorithm 2. The construction procedure of atom coordinates are discussed in Appendix A. The definition of evaluation metrics are listed in Appendix E. The source code of this work is available at https://github.com/lujiarui/Str2Str.

ACKNOWLEDGMENTS

We thank Zhaocheng Zhu and Sophie Xhonneux for helpful feedback as well as anonymous reviewers for their constructive suggestion and comments. This project is supported by Twitter, Intel, the Natural Sciences and Engineering Research Council (NSERC) Discovery Grant, the Canada CIFAR AI Chair Program, Samsung Electronics Co., Ltd., Amazon Faculty Research Award, Tencent AI Lab Rhino-Bird Gift Fund, an NRC Collaborative R&D Project (AI4D-CORE-06) as well as the IVADO Fundamental Research Project grant PRF-2019-3583139727.

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

## A CONSTRUCTION OF RESIDUES COORDINATES

### A.1 TRANSFORMATION BETWEEN FRAMES AND EUCLIDEAN COORDINATES

As discussed in Section 2, the atom coordinates on protein backbone are parameterized by the frames and can be constructed by applying the transformation induced by frames to the local coordinates with the idealized bond lengths and angles. The local coordinates sets, depending on the amino acid type, were used in Jumper et al. (2021) and originally introduced in Engh & Huber (2012). For example, the corresponding coordinates for amino acid ALA are listed as below, with the set centered with respect to $C^\alpha$ as origin:

$$\boldsymbol{x}^{\mathrm{N}} : (-0.525, 1.363, 0.000)$$
$$\boldsymbol{x}^{\mathrm{C}^\alpha} : (0.000, 0.000, 0.000)$$
$$\boldsymbol{x}^{\mathrm{C}} : (1.526, -0.000, -0.000)$$
$$\boldsymbol{x}^{\mathrm{C}^\beta} : (-0.529, -0.774, -1.205)$$
$$\boldsymbol{x}^{\mathrm{O}} : (0.627, 1.062, 0.000)$$

The local-to-global procedure for any atom coordinates can be performed by applying the transformation induced by a backbone frame $T^{\mathrm{frame}} = (R^{\mathrm{frame}}, \boldsymbol{v}^{\mathrm{frame}})$ to the local coordinates:

$$\boldsymbol{x}_{global} = T^{\mathrm{frame}} \circ \boldsymbol{x}_{\mathrm{local}} = R^{\mathrm{frame}} \boldsymbol{x}_{\mathrm{local}} + \boldsymbol{v}^{\mathrm{frame}}, \tag{9}$$

where $R^{\mathrm{frame}} \in \mathrm{SO}(3)$ is a $3 \times 3$ rotation matrix while $\boldsymbol{v}^{\mathrm{frame}} \in \mathbb{R}^3$ is a translation vector.

As in Jumper et al. (2021), the backbone oxygen is additionally parameterized by the torsion angle $\psi$, which is based on the axis-rotation of $C^\alpha$-C single bond. The oxygen coordinates can be obtained by applying torsion transformation followed by the local-to-global transformation similar to other backbone atoms (for $i$-th residue as example):

$$\boldsymbol{x}_{global}^{\mathrm{O}} = T^{\mathrm{frame}} \circ T^\psi \circ \boldsymbol{x}_{\mathrm{local}}^{\mathrm{O}}, \tag{10}$$

where the $T^{\mathrm{frame}} := (R, \boldsymbol{v})$ indicates the local-to-global transformation defined in Section 2 while $T^\psi := (R^\psi, \boldsymbol{v}^\psi)$ is the additional transformation induced by $\psi$. Let $[\sin(\psi), \cos(\psi)]$ as its sin-cos representation, the corresponding transformation can be write in:

$$R^\psi = \begin{bmatrix} 1 & 0 & 0 \\ 0 & \sin(\psi) & -\cos(\psi) \\ 0 & \cos(\psi) & \sin(\psi) \end{bmatrix}, \boldsymbol{v}^\psi = \boldsymbol{x}^{\mathrm{C}}. \tag{11}$$

To construct the frames from coordinates of protein structures in PDB, we adopt the Gram-Schmidt process in Jumper et al. (2021) from the coordinates of $N$-$C^\alpha$-$C$ for each residue, which is:

---

**Algorithm 1** Frame from three points (Algo 21, Jumper et al. (2021)

1: **Require:** $\boldsymbol{x}_1, \boldsymbol{x}_2, \boldsymbol{x}_3$ as global coordinates of atoms N, CA, C.
2: $\boldsymbol{u}_1 = \boldsymbol{x}_3 - \boldsymbol{x}_2$
3: $\boldsymbol{u}_2 = \boldsymbol{x}_1 - \boldsymbol{x}_2$
4: $\boldsymbol{e}_1 = \boldsymbol{u}_1 / \|\boldsymbol{u}_1\|$
5: $\boldsymbol{e}_2 = \frac{\boldsymbol{u}_2 - \boldsymbol{e}_1(\boldsymbol{e}_1^\top \boldsymbol{u}_2)}{\|\boldsymbol{u}_2 - \boldsymbol{e}_1(\boldsymbol{e}_1^\top \boldsymbol{u}_2)\|}$
6: $\boldsymbol{e}_3 = \boldsymbol{e}_1 \times \boldsymbol{e}_2$
7: $R = \mathrm{CONCAT}(\boldsymbol{e}_1, \boldsymbol{e}_2, \boldsymbol{e}_3)$
8: $v = \boldsymbol{x}_2$
9: **return** $R, v$

---

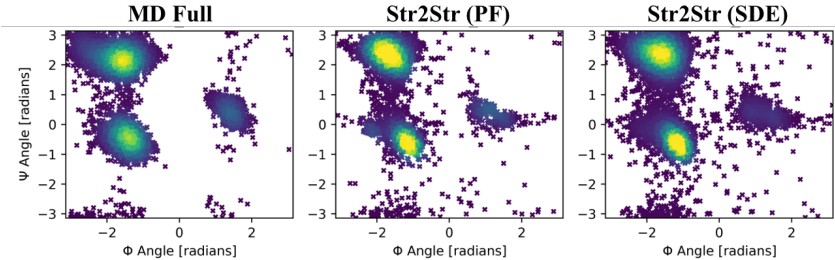

Figure S1: The Ramachandran plot of $\phi, \psi$ angles for the protein Chignolin as an example. STR2STR(PF and SDE) are compared with the snapshots of the reference full MD simulation.

## A.2 RIGID GROUPS OF AMINO ACIDS

According to the representation used in the main text, we here provide the detailed rigid group definition to construct the corresponding Euclidean atom coordinates. The notations in this table are aligned with the settings in Jumper et al. (2021).

Table S1: Specification of all the rigid groups for each residue type.

| AA type | Backbone | $\psi$ | $\chi_1$ | $\chi_2$ | $\chi_3$ | $\chi_4$ |
|---|---|---|---|---|---|---|
| **ALA** | N, $C^\alpha$, C, $C^\beta$ | O | - | - | - | - |
| **ARG** | N, $C^\alpha$, C, $C^\beta$ | O | $C^\gamma$ | $C^\delta$ | $N^\epsilon$ | $N^{\eta 1}, N^{\eta 2}, C^\zeta$ |
| **ASN** | N, $C^\alpha$, C, $C^\beta$ | O | $C^\gamma$ | $N^{\delta 2}, O^{\delta 1}$ | - | - |
| **ASP** | N, $C^\alpha$, C, $C^\beta$ | O | $C^\gamma$ | $O^{\delta 1}, O^{\delta 2}$ | - | - |
| **CYS** | N, $C^\alpha$, C, $C^\beta$ | O | $S^\gamma$ | - | - | - |
| **GLN** | N, $C^\alpha$, C, $C^\beta$ | O | $C^\gamma$ | $C^\delta$ | $N^{\epsilon 2}, O^{\epsilon 1}$ | - |
| **GLU** | N, $C^\alpha$, C, $C^\beta$ | O | $C^\gamma$ | $C^\delta$ | $O^{\epsilon 1}, O^{\epsilon 2}$ | - |
| **GLY** | N, $C^\alpha$, C | O | - | - | - | |
| **HIS** | N, $C^\alpha$, C, $C^\beta$ | O | $C^\gamma$ | $C^{\delta 2}, N^{\delta 1}, C^{\epsilon 1}, N^{\epsilon 2}$ | - | - |
| **ILE** | N, $C^\alpha$, C, $C^\beta$ | O | $C^{\gamma 1}, C^{\gamma 2}$ | $C^{\delta 1}$ | - | - |
| **LEU** | N, $C^\alpha$, C, $C^\beta$ | O | $C^\gamma$ | $C^{\delta 1}, C^{\delta 2}$ | - | - |
| **LYS** | N, $C^\alpha$, C, $C^\beta$ | O | $C^\gamma$ | $C^\delta$ | $C^\epsilon$ | $N^\zeta$ |
| **MET** | N, $C^\alpha$, C, $C^\beta$ | O | $C^\gamma$ | $S^\delta$ | $C^\epsilon$ | - |
| **PHE** | N, $C^\alpha$, C, $C^\beta$ | O | $C^\gamma$ | $C^{\delta 1}, C^{\delta 2}, C^{\epsilon 1}, C^{\epsilon 2}, C^\zeta$ | - | - |
| **PRO** | N, $C^\alpha$, C, $C^\beta$ | O | $C^\gamma$ | $C^\delta$ | - | - |
| **SER** | N, $C^\alpha$, C, $C^\beta$ | O | $O^\gamma$ | - | - | - |
| **THR** | N, $C^\alpha$, C, $C^\beta$ | O | $C^{\gamma 2}, O^{\gamma 1}$ | - | - | - |
| **TRP** | N, $C^\alpha$, C, $C^\beta$ | O | $C^\gamma$ | $C^{\delta 1}, C^{\delta 2}, C^{\epsilon 2}, C^{\epsilon 3}, N^{\epsilon 2}, C^{\eta 2}, C^{\zeta 2}, C^{\zeta 3}$ | - | - |
| **TYR** | N, $C^\alpha$, C, $C^\beta$ | O | $C^\gamma$ | $C^{\delta 1}, C^{\delta 2}, C^{\epsilon 1}, C^{\epsilon 2}, O^\eta, C^\zeta$ | - | - |
| **VAL** | N, $C^\alpha$, C, $C^\beta$ | O | $C^{\gamma 1}, C^{\gamma 2}$ | - | - | - |

## A.3 CHIRALITY

To examine whether the STR2STR can generate correct chirality with respect to the input protein, we showcase the Ramachandran plots for torsion angles in conformation from STR2STR (PF and SDE) and MD trajectory, where we select as example the smallest peptide Chignolin (length=10) in fast folding targets. The results are shown in Figure S1. In this case, there are three clusters in the $\phi - \psi$ within the 2D space $[-\pi, \pi]^2$ and STR2STR has basically covered three of them. The correct capturing of chirality comes from the backbone construction procedure in the same manner as AlphaFold2 (Jumper et al., 2021).

## B EQUIVARIANT STRUCTURE-TO-STRUCTURE TRANSLATION

To better model particle system such as protein conformation, equivariance with respect to spatial transformations (eg. rotation $90°$ along some axis) should be injected as an inductive bias for better generalization. For protein conformations, chirality is important to convey structure-function relationship and should be conserved during the mapping. Therefore, we consider only global translation

and rotation elements (or roto-translation operations), excluding the reflection which can change the chirality. Especially, the conformations exist in 3D space and thus realize the translation vector $\boldsymbol{v} \in \mathbb{R}^3$ and rotation matrix $R \in \mathbb{R}^{3\times3}$. Under such constraints, the function (mapping) is said to be SE(3)-equivariant. The pseudo-code for the forward-backward process is shown in Algorithm 2 for better illustration.

---

**Algorithm 2** Forward-backward process of protein frames

---

1: **Require:** input frames $\mathbf{T}$, scheduled perturbation scales $\{T_{\delta_i}\}_{k=1}^K$; score network $\mathbf{s}_\theta$; the minimum of diffusion time $\epsilon > 0$, noise scale factor $\zeta > 0$.
2: $S_\mathbf{T} = \{\}$    *// initialize result set*
3: **for** $k = 1$ to $K$ **do**
4:      $\mathbf{T}^{(0)} \leftarrow \mathbf{T}$    *// initialize state*
5:      $\{t_1, \ldots, t_M\} \leftarrow \text{Discretize}([\epsilon, T_{\delta_k}])$    *// discretize time domain*
6:      $\tau \leftarrow (T_{\delta_k} - \epsilon)/M$
7:      *// forward perturbation*
8:      $(T_1^{(0)}, \ldots, T_n^{(0)}) \leftarrow \mathbf{T}^{(0)}$
9:      **for** $i = 1$ to $n$ **do**
10:          $(R_i, v_i) \leftarrow T_i^{(0)}$
11:          $R_i^{(t_M)} \sim p_{t_M|0}^{\text{rot}}(R_i^{(t_M)}|R_i), \; v_i^{(t_M)} \sim p_{t_M|0}^{\text{trans}}(v_i^{(t_M)}|v_i)$    *// forward diffusion*
12:          $T_i^{(t_M)} \leftarrow (R_i^{(t_M)}, v_i^{(t_M)})$
13:      $\mathbf{T}^{(t_M)} \leftarrow (T_1^{(t_M)}, \ldots, T_n^{(t_M)})$
14:      *// backward annealing*
15:      **for** $j = M - 1, \ldots, 1$ **do**
16:          $(T_1^{(t_{j+1})}, \ldots, T_n^{(t_{j+1})}) \leftarrow \mathbf{T}^{(t_{j+1})}$
17:          $(s_1^{(t_{j+1})}, \ldots, s_n^{(t_{j+1})}) \leftarrow \mathbf{s}_\theta(\mathbf{T}^{(t_{j+1})}, t_{j+1})$    *// estimated scores*
18:          **for** $i = 1$ to $n$ **do**
19:              $(R_i, v_i) \leftarrow T_i^{(t_j)}$
20:              $(s_i^R, s_i^v) \leftarrow s_i^{(t_{j+1})}$    *// disentangle scores*
21:              $z_v \sim \mathcal{N}_{\mathbb{R}^3}(\mathbf{0}, \mathbf{I})$
22:              $z_R \sim \mathcal{TN}_{R_i}(\mathbf{0}, \mathbf{I})$    *// tangent space of rotation*
23:              $v_i^* \leftarrow \boldsymbol{P}[\frac{1}{2}v_i + s_i^v] + \zeta\sqrt{\tau}z_v$ *// translation update w/ removing center of mass*
24:              $R_i^* \leftarrow s_i^R + \zeta\sqrt{\tau}z_R$ *// rotation update*
25:              $T_i^{(t_j)} \leftarrow \exp_{T_i^{(t_{j+1})}}[(R_i^*, v_i^*)]$    *// exponential map*
26:          $\mathbf{T}^{(t_j)} \leftarrow (T_1^{(t_j)}, \ldots, T_n^{(t_j)})$
27:          $S_\mathbf{T} = S_\mathbf{T} \cup \{\mathbf{T}^{(\epsilon)}\}$
28: **return** $S_\mathbf{T}$

---

**Invariant Backbone Frame Update.** The update rule of frames $\mathbf{T}$ follows the implementation of Algo. 23 in Jumper et al. (2021) which can be viewed as a composition between original frame and the frame update induced by the single representation $s_i$. The roto-translation equivariant property of frame update (Algorithm 3) is straightforward: the resulting updated frames from each layer are simply a (right) composition of the update $T_i^{\text{update}}$, which is induced by the invariant single representation $\{s_i\}$ and thus equivariant. In each layer, the IPA only takes invariant features of the input frames to update the single representation $s_i$, which is, as being named, invariant to global roto-translations on the atom coordinates (backbone frames). As a remark, the mentioned addition operations ("+") for frames (such as in Eq. (4)) symbolically denotes the frame update as the composition of two frames $T_1 + T_2 \equiv T_1 \circ T_2 = (R_1, \boldsymbol{v}_1) \circ (R_2, \boldsymbol{v}_2) = (R_1 R_2, R_1 \boldsymbol{v}_2 + \boldsymbol{v}_1)$.

**Proposition 1.** *Proof.* Without loss of generality, consider the residue $i$ with atoms coordinates $\mathbf{x}_{0,i}$ and backbone frame $T_{0,i}$ where subscript 0 indicates the input. The corresponding frame $T_i = (R_i, \boldsymbol{v}_i)$ is sampled via the forward-backward (FB) process in Eq. 4. For the forward component, the perturbation kernel (drift, diffusion coefficients) defined in Eq. 2 has no dependency on the input frames and thus the update of forward is invariant to global roto-translation. Therefore, the

---

**Algorithm 3** Backbone frame update

---

1: **Require:** input frame $T_i^{(l)}$, updated single representation $s_i^{(l+1)}$ of $i$-th residue for layer $l$.
2: $b_i, c_i, d_i, \boldsymbol{v}_i \leftarrow \text{Linear}(s_i^{(l+1)})$   $//b_i, c_i, d_i \in \mathbb{R}, v_i \in \mathbb{R}^3$
3: $(a_i, b_i, c_i, d_i) \leftarrow (1, b_i, c_i, d_i)/\|(1, b_i, c_i, d_i)\|$   // normalize quaternion
4: $R_i = \text{Quat\_to\_rotmat}(a_i, b_i, c_i, d_i)$   // convert normalized quaternion to rotation matrix
5: $T_i^{\text{update}} \leftarrow (R_i, \boldsymbol{v}_i)$
6: $T_i^{(l+1)} \leftarrow T_i \circ T_i^{\text{update}}$
7: **return** $T_i^{(l+1)}$

---

composition of input $T_{0,i}$ and the forward update is equivariant; on the other hand, the backward component is similar to that of forward, with the only difference on the drift term as the score function. Since the score functions are assumed to be equivariant to global roto-translation, the backward process also assumes equivariant property. By definition of equivariance, we can extract the global roto-translation transformation $T_{global}$ from the output frames by:

$$\text{FB}_T(T_{global} \circ T_{0,i}) = T_{global} \circ \text{FB}_T(T_{0,i}), \tag{12}$$

where $\text{FB}_T(\cdot)$ indicates the forward-backward process for frames defined in Eq. 4, With equivariant transformed frames $T_i = (R_i, \boldsymbol{v}_i)$, the atom coordinates constructed from local-to-global transformation can also be equivariant. $\forall \boldsymbol{x}_{\text{global}} \in \mathbb{R}^3$, let $\text{FB}_x(\cdot)$ be the corresponding FB-operator on global coordinates instead of frames:

$$
\begin{aligned}
&\text{FB}_x\left(T_{global} \circ \boldsymbol{x}_{\text{global}}\right) \\
&= \text{FB}_x\left[T_{global} \circ (T_{0,i} \circ \boldsymbol{x}_{\text{local}})\right] \quad \textit{(definition of global coordinates)} \\
&= \text{FB}_x\left[(T_{global} \circ T_{0,i}) \circ \boldsymbol{x}_{\text{local}}\right] \quad \textit{(associative property of transformation)} \\
&= T_{global} \circ \text{FB}_T(T_{0,i}) \circ \boldsymbol{x}_{\text{local}} \quad \textit{(equivariance on frames)} \\
&= T_{global} \circ \text{FB}_x(T_{0,i} \circ \boldsymbol{x}_{\text{local}}) \quad \textit{(definition of operator)} \\
&= T_{global} \circ \text{FB}_x(\boldsymbol{x}_{\text{global}}), 
\end{aligned}
\tag{13}
$$

which indicates equivariant property for any atom coordinates $\boldsymbol{x}_{\text{global}}$ on backbone. Plus the invariant nature of torsion angles as internal coordinates, the rest of coordinates is thus equivariant with all backbone coordinates being equivariant. $\square$

## C   ABLATION STUDIES

### C.1   TRAINING WITH DIFFERENT COIL CUTOFF

The sampling performance by learning from the amortized objective in Section 3.3 relies on the structural characteristics of the training dataset. Here, we studied how different ratio of structural regions in the training structures would affect the benchmarking performance. In specific, we screened out the structures in the training set according to the ratio of coil (no secondary structure presents). To construct the training set, we totally set three coil ladders $r_c$: 0.5, 0.75, 1.0, where, for example, $r_c = 0.5$ indicates the maximal number of residues that forms the coil region is less than or equal to $0.5 * L$ ($L$ is the number of residues of such protein structure). To annotate the secondary structure to records in PDB, we utilized the DSSP algorithm (Kabsch & Sander, 1983). Results are shown in Table S2. For the result in the main text, the $r_c = 0.5$ is adopted for generally better structural validity. However, we found that an intermediate cutoff with $r_c = 0.75$ can have better performance for distributional matching.

### C.2   ALTERNATIVE NOISE DURING INFERENCE

The forward-backward process proposed in Section 3.2 allows in-distribution perturbed samples $\mathbf{x}_t$ during inference by using the same perturbation kernel as training. As an ablation, we alternatively experimented the inference by simply replaced the Gaussian noise parameterized by $(\mu, \sigma^2)$ during

Table S2: Benchmark results of fast folding proteins (Lindorff-Larsen et al., 2011) for different filtering ratio of coil used for training. The best result for each metric is **bolded**.

| Setting | Val-Clash(↑) | Val-Bond(↑) | JS-PwD(↓) | JS-TIC(↓) | JS-Rg (↓) | MAE-TM(↓) | MAE-RMSD(↓) |
|---|---|---|---|---|---|---|---|
| $r_c = 0.5$ (PF) | 0.963 | **0.992** | 0.375 | 0.397 | 0.448 | 0.150 | 0.209 |
| $r_c = 0.75$ (PF) | 0.953 | 0.978 | 0.352 | **0.373** | 0.429 | 0.137 | 0.202 |
| $r_c = 1.0$ (PF) | 0.927 | 0.919 | 0.360 | 0.393 | 0.471 | 0.141 | 0.201 |
| $r_c = 0.5$ (SDE) | **0.977** | 0.982 | 0.348 | 0.400 | 0.365 | **0.133** | **0.184** |
| $r_c = 0.75$ (SDE) | 0.972 | 0.933 | **0.323** | 0.384 | **0.348** | 0.143 | 0.205 |
| $r_c = 1.0$ (SDE) | 0.931 | 0.909 | 0.332 | 0.394 | 0.389 | 0.143 | 0.204 |

forward perturbation with the corresponding uniform noise within the compact region of $[\mu-3\sigma, \mu+3\sigma]$, where $\mu$ is mean and $\sigma$ is the standard deviation of Gaussian. Although the support of such uniform distribution is quite similar to the original Gaussian ($3\sigma$ rule: the span above has covered 99.73% area), the distributional characteristics of them can be quite different. As shown in Table S3, the use of uniform perturbation kernel does not significantly change the performance of STR2STR. This result aligns with the previous claim that forward perturbation mainly does exploration and the backward annealing executes exploitation in the forward-backward process. Different exploration scheme induced by the random noise can be studied as future works.

Table S3: Benchmark results of fast folding proteins (Lindorff-Larsen et al., 2011) by using different perturbation distribution during inference. The best result for each metric is **bolded**.

| Setting | Val-Clash(↑) | Val-Bond(↑) | JS-PwD(↓) | JS-TIC(↓) | JS-Rg (↓) | MAE-TM(↓) | MAE-RMSD(↓) |
|---|---|---|---|---|---|---|---|
| Gaussian (PF) | 0.963 | **0.992** | 0.375 | 0.397 | 0.448 | 0.150 | 0.209 |
| Gaussian (SDE) | **0.977** | 0.982 | 0.348 | 0.400 | 0.365 | **0.133** | **0.184** |
| $3\sigma$-Uniform (PF) | 0.962 | **0.992** | 0.360 | 0.397 | 0.413 | 0.156 | 0.231 |
| $3\sigma$-Uniform (SDE) | 0.976 | 0.983 | **0.343** | **0.395** | **0.356** | 0.141 | 0.192 |

## C.3 THE CHOICE OF FOLDING MODELS

To sampling from the conformation space, STR2STR requires an initial structure input to provide necessary information of the target protein. Thanks to the accurate performance of single structure folding models (Jumper et al., 2021; Lin et al., 2023), this is a tractable assumption and no simulation data is introduced. To shown the effect of different folding module on STR2STR, we benchmarked both structure prediction models: single sequence-based ESMFold and MSA-based model AlphaFold2 and results are shown in Table S4. Both models are executed under default settings. Note that the structure prediction accuracy of ESMFold can be worse than AlphaFold2 with lower plDDT. The results demonstrate that the sampling of STR2STR is basically robust to the choice of folding models. In our work, we mainly used ESMFold to obtain the initial point because its inference only depends on a single input sequence, which is fast and aligned with other baselines.

Table S4: Benchmark results of fast folding proteins (Lindorff-Larsen et al., 2011) by using different folding modules to initialize conformation for sampling. The best result for each metric is **bolded**.

| Setting | Val-Clash(↑) | Val-Bond(↑) | JS-PwD(↓) | JS-TIC(↓) | JS-Rg (↓) | MAE-TM(↓) | MAE-RMSD(↓) |
|---|---|---|---|---|---|---|---|
| ESMFold (PF) | 0.963 | 0.992 | 0.375 | 0.397 | 0.448 | 0.150 | 0.209 |
| ESMFold (SDE) | **0.977** | 0.984 | 0.348 | 0.400 | 0.365 | **0.133** | **0.184** |
| AlphaFold2 (PF) | 0.964 | **0.993** | 0.365 | **0.374** | 0.453 | 0.148 | 0.229 |
| AlphaFold2 (SDE) | **0.977** | 0.981 | **0.336** | 0.376 | **0.351** | **0.133** | 0.201 |

## C.4 ABLATING THE ORIENTATION MODELING

To illuminate whether the modeling of orientations (say rotation matrix) of frame is necessary, we conduct an ablation study by canceling out the rotation modeling and only modeling the $C^\alpha$. Correspondingly, the ablated model is trained w/o rotation loss. To craft the rotations for IPA input during inference, we leverages two strategies that either use: (1) input rotations or (2) random rotations for

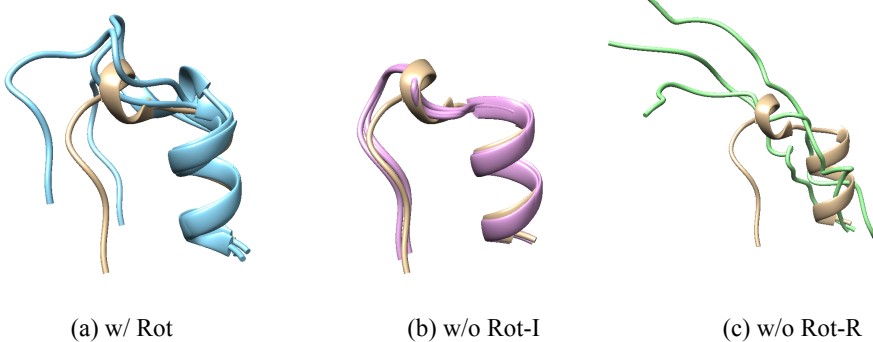

|(a) w/ Rot | (b) w/o Rot-I | (c) w/o Rot-R |

Figure S2: Visualization of Trp-cage samples of STR2STR for rotation ablation, with (a) rotation modeled; (b) no rotation modeled but using input rotations; and (c) no rotation modeled but using random rotations. Across three subfigures, the input (folded) conformation is colored golden, on which the samples of each setting are aligned and superposed.

each backbone frame. As shown in Table S5, the model w/o rotation loss experiences a significant performance drop. To illustrate such gap, we take for example the Trp-cage and visualize the structures in Figure S2. It demonstrates that: (1) when using the input rotation, the model almost loses diversity but only generate conformations close to the input; (2) when using random rotations, the model fails to capture the local secondary structures but generates random coil. It implies orientation modeling can be necessary for the current STR2STR framework.

Table S5: Benchmark results of fast folding proteins (Lindorff-Larsen et al., 2011) by ablating orientation modeling. Two strategies are used to construct the (pseudo) rotations during inference: (1) *w/o Rot-I* that uses the input rotations; and (2) *w/o Rot-R* that adopts randomly sampled rotations. The best result for each metric is **bolded**.

| Setting | Val-Clash(↑) | Val-Bond(↑) | JS-PwD(↓) | JS-TIC(↓) | JS-Rg (↓) | MAE-TM(↓) | MAE-RMSD(↓) |
|---|---|---|---|---|---|---|---|
| w/ Rot (PF) | 0.963 | 0.992 | 0.375 | **0.397** | 0.448 | 0.150 | 0.209 |
| w/o Rot-I (PF) | **1.000** | **1.000** | 0.675 | 0.619 | 0.612 | 0.643 | 0.748 |
| w/o Rot-R (PF) | 0.002 | 0.102 | 0.378 | 0.446 | 0.459 | 0.191 | 0.335 |
| w/ Rot (SDE) | 0.977 | 0.982 | **0.348** | 0.400 | **0.365** | **0.133** | **0.184** |
| w/o Rot-I (SDE) | **1.000** | **1.000** | 0.639 | 0.589 | 0.5951 | 0.193 | 0.486 |
| w/o Rot-R (SDE) | 0.001 | 0.105 | 0.427 | 0.442 | 0.585 | 0.612 | 0.735 |

## C.5 ADJUSTING NOISE SCALE FOR SDE

Similar to the temperature factor for sampling from categorical distributions, the sampling can be affected by simply re-scaling the scale of noise during backward process. Several previous works (Ingraham et al., 2022; Watson et al., 2022; Yim et al., 2023) applied such strategies to the task of backbone design to generate backbones with controlled diversity and designability. For the zero-shot conformation sampling, we also investigated the effect of different noise scale on the ensemble metrics. As shown in Table S6, we found that although lower noise scale can improve the sampling validity, adjusting (increase or decrease from 1.0) noise scale can hurt the sampling fidelity for SDE. Improving both the fidelity and diversity of STR2STR by scheduling the noise scales during backwards can be an interesting future work.

## D IMPLEMENTATION DETAILS

**Definition of Auxiliary Losses.** The auxiliary losses are composed of backbone mean square error (MSE) $\mathcal{L}_{\text{backb}}$ and distogram loss $\mathcal{L}_{\text{dist}}$. Firstly, the backbone MSE loss for a structure with $n$

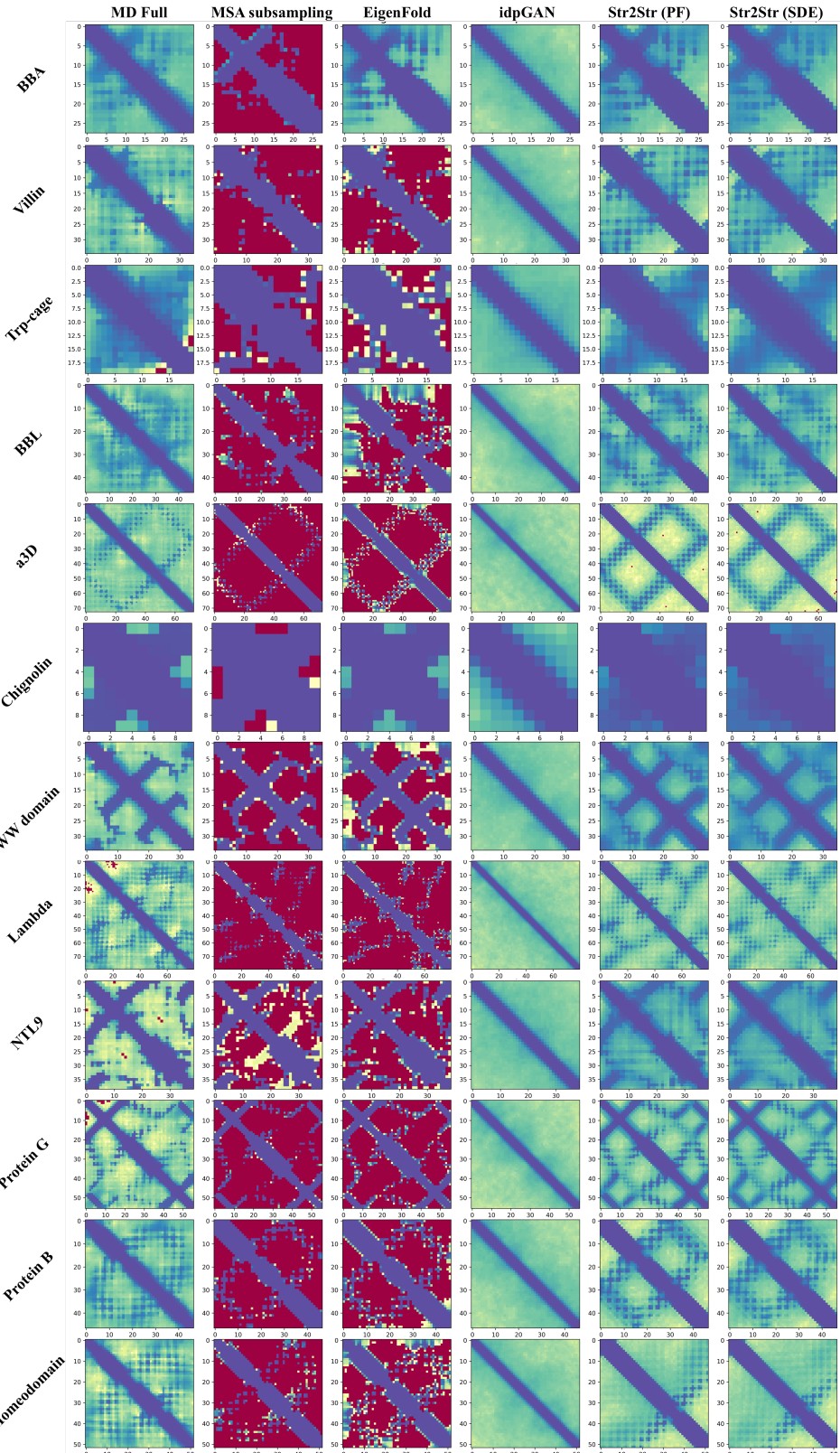

Figure S3: Contact map for each target protein in the fast folding benchmarks. Samples from reference MD full, baselines as well as STR2STR are shown.

Table S6: Benchmark results of fast folding proteins (Lindorff-Larsen et al., 2011) for different scale of noise during backward diffusion for SDE. The best result for each metric is **bolded**.

| Setting | Val-Clash($\uparrow$) | Val-Bond($\uparrow$) | JS-PwD($\downarrow$) | JS-TIC($\downarrow$) | JS-Rg ($\downarrow$) | MAE-TM($\downarrow$) | MAE-RMSD($\downarrow$) |
|---|---|---|---|---|---|---|---|
| Noise scale = 0.1 | **0.985** | **0.992** | 0.399 | 0.419 | 0.420 | 0.156 | 0.214 |
| Noise scale = 0.25 | **0.985** | **0.992** | 0.393 | 0.419 | 0.414 | 0.152 | 0.207 |
| Noise scale = 0.5 | 0.984 | 0.990 | 0.382 | 0.409 | 0.403 | 0.147 | 0.204 |
| Noise scale = 0.75 | **0.985** | 0.986 | 0.367 | 0.404 | 0.386 | 0.142 | 0.186 |
| Noise scale = 1.0 | 0.979 | 0.981 | **0.349** | **0.400** | **0.362** | **0.133** | **0.184** |
| Noise scale = 2.0 | 0.668 | 0.722 | 0.371 | 0.457 | 0.457 | 0.186 | 0.474 |
| Noise scale = 5.0 | 0.434 | 0.000 | 0.585 | 0.587 | 0.798 | 0.285 | 3.043 |

residues in training set is defined as:

$$\mathcal{L}_{\text{backb}} = \frac{1}{4n} \sum_{i=1}^{n} \sum_{t \in \{N, C^\alpha, C, O\}} \|\boldsymbol{x}_i^t - \hat{\boldsymbol{x}}_i^t\|^2, \tag{14}$$

where $\hat{\boldsymbol{x}}_i^t$ is the predicted coordinates for the atom $t$ of residue $i$ by applying the local-to-global coordinate construction by the predicted frames, and the summation is iterated over the residues $i = 1, 2, \ldots, n$ and the types of backbone atoms shared among all types of amino acids, say $\{N, C^\alpha, C, O\}$. For the distogram loss, let $d_{i,j}^{t_1, t_2}$ be the atomic distance between atom $t_1$ and $t_2$ of residue $i$ and $j$ respectively, the loss is a penalty to the distance error of local contacting neighbors within 6Å:

$$\mathcal{L}_{\text{dist}} = \frac{\sum_{i,j=1}^{n} \sum_{t_1, t_2 \in \{N, C^\alpha, C, O\}} \|d_{i,j}^{t_1, t_2} - \hat{d}_{i,j}^{t_1, t_2}\|^2 \cdot \mathbf{1}\{d_{i,j}^{t_1, t_2} < 6\text{Å}, t_1 \neq t_2\}}{\sum_{i,j=1}^{n} \sum_{t_1, t_2 \in \{N, C^\alpha, C, O\}} \mathbf{1}\{d_{i,j}^{t_1, t_2} < 6\text{Å}, t_1 \neq t_2\}} \tag{15}$$

As indicated in Section 3.3, the overall training loss is $\mathcal{L} = \mathcal{L}_{\text{dsm}} + \alpha \mathcal{L}_{\text{backb}} + \beta \mathcal{L}_{\text{dist}}$ $(\alpha, \beta > 0)$. Here, we use $\alpha = \beta = 0.25$ to combine auxiliary loss with the denoising score matching loss. Following Yim et al. (2023), the auxiliary loss is only applied for diffusion steps close to $t = 0$, which is specifically when $t < T/4$.

**Score Predictions.** The DenoisingIPA will finally output an updated frame $\mathbf{T}^*$ from the noisy input $\mathbf{T}^t$ at time $t$. According to the SE(3)$^n$ diffusion (Yim et al., 2023), the corresponding score constructed from such output for the $i$-th residue frame $T_i^t = (R_i^t, v_i^t)$ is:

$$s_\theta(T_i^t, t) = \left[ \frac{R_i^t}{\omega} \log(R_i^*) \frac{\partial}{\partial \omega} p_{t|0}(R_i^t | R_i^*), -\frac{v_i^t - e^{-\frac{1}{2}\beta(t)} v_i^*}{1 - e^{-\beta(t)}} \right], \tag{16}$$

where $\omega = \text{Axis\_angle}(R_i^{*\top} R_t)$ and $p_{t|0}(\cdot | R)$ is defined in Eq. 6. $\log(R_i^*)$ is the logarithm mapping defined by $\log R = \frac{\theta}{2 \sin \theta}(R^\top - R)$ with $1 + 2 \cos \theta = \text{trace}(R)$ (Leach et al., 2022).

**Training Diffusion Modules.** For the diffusion on translation components $\in \mathbb{R}^3$, we use the linear beta schedule of VPSDE (Song et al., 2020) with $\beta_{\min} = 0.1$ and $\beta_{\max} = 20$, say $\beta(t) = \beta_{\min} + (t/T)(\beta_{\max} - \beta_{\min})$; for the rotation diffusion, we use the logarithm sigma schedule of VESDE as $\sigma(t) = \log(t \cdot e^{\sigma_{\max}} + (T - t) \cdot e^{\sigma_{\min}})$ with $\sigma_{\min} = 0.1$ and $\sigma_{\max} = 1.5$. In the score matching objective, the maximal time scale is selected to be $\tau_m = T = 1.0$. To optimize the parameters of DenoisingIPA, the Adam optimizer (Kingma & Ba, 2014) is used with lr $= 10^{-4}$ and $\beta_1 = 0.9, \beta_2 = 0.999$. The training time is approximately $\sim$30 GPU days on the NVIDIA Tesla V100-SXM2-32GB GPU.

**Training Data.** To curate the training data for STR2STR, we collected the mmCIF structures from the Protein Data Bank (PDB) on June 9th, 2023 in single chain (monomer) with experimental resolution better than 5Å. Then we screened out those records with length $L > 512$ or $L < 10$, which prevents memory overflow on a single GPU while removing too short peptides. To avoid possible data leakage, the PDB records in the training set with the same sequence identity as test proteins were removed. After filtering, there are totally 26,459 monomeric structures in the training set.

| Parameter | Number |
|---|---|
| Single channel $c_s$ | 256 |
| Pair channel $c_z$ | 128 |
| Initial skip channel $c_{\text{skip}}$ | 64 |
| Number of heads $n_{\text{head}}$ | 8 |
| Number of query-key points $n_{qk}$ | 8 |
| Number of value points $n_v$ | 12 |
| Number of MHA heads $n_{\text{MHA}_{\text{head}}}$ | 4 |
| Number of MHA layers $L_{\text{MHA}}$ | 2 |
| Number of IPA layers $L_{\text{IPA}}$ | 4 |

Table S7: Hyperparameters of DenoisingIPA module.

**Hyperparameters of DenoisingIPA.** The definition of hyperparameters for DenoisingIPA networks is shown in Table S7.

**Test Targets.** The fast folding targets with the corresponding PDB structures are listed as follows (Lindorff-Larsen et al., 2011): Chignolin (No PDB entry, reported in the supplementary of Honda et al. (2008)), Trp-cage (PDB entry 2JOF), BBA (PDB entry 1FME), Villin (PDB entry 2F4K), WW domain (PDB entry 2F21), NTL9 (PDB entry NTL9), BBL (PDB entry 2WXC), Protein B (PDB entry 1PRB), Homeodomain (PDB entry 2P6J), Protein G (PDB entry 1MIO), $\alpha$3D (PDB entry 2A3D) and Lambda-repressor (PDB entry 1LMB). The protein BPTI has the corresponding PDB deposit with PDB id 5PTI. The reference MD trajectories for evaluation were obtained by sending requests to the authors of Lindorff-Larsen et al. (2011) and Shaw et al. (2010).

**Inference Stage.** During inference, we discretized the time domain $[\epsilon, 1]$ with 1,000 timesteps to reverse diffusion process via Langevin Dynamics. In practice, a minimum diffusion time $\epsilon = 0.01$ instead of zero is used for numerical stability. To obtain initial conformation from folding models, we utilized both ESMFold (Lin et al., 2023) and AlphaFold2 (Jumper et al., 2021) in Appendix C.3 to obtain the initial structure for conformation sampling of STR2STR from a sequence input. For each test target, we ran the inference procedure of pretrained ESMFold-v1 using the default setting and ran the ColabFold (Mirdita et al., 2022) implementation of AlphaFold2 with accelerated MSA search. Among predictions, the model with the highest confidence score (plDDT) is adopted to initialize the structure-to-structure translation. Since different perturbation scale, i.e. transition times $T_\delta$ can lead to different scale of exploration, we applied a linear schedule of $T_\delta$ between $[t_{\min}, t_{\max}]$. For fast folding proteins, $t_{\min} = 0.25$, $t_{\max} = 0.7$ and the stride is 0.05. For the structural protein BPTI, $t_{\min} = 0.10$, $t_{\max} = 0.15$ and the stride 0.05 are used; moreover, we anneal the noise scale to be 0.1 for the SDE sampling for BPTI. The resulting ensemble of STR2STR is obtained by merging the sampled conformations from each perturbation scale and then evaluated.

**Baselines.** The conformations sampled from MSA subsampling (Del Alamo et al., 2022) involves limited recycle and depth-reduced multiple sequence alignments (MSA) during AlphaFold2 (AF2) inference. Specifically, JackHMMER and HHBlits were used for searching MSA from databases including UniRef90, MGnify, and BFD, following AF2's original pipeline (Jumper et al., 2021) instead of using MMSeqs2 (Steinegger & Söding, 2017) as in Del Alamo et al. (2022). The number of recycles was set to one. Only a small portion of the found MSA was used as input to compute the MSA representation fed to Evoformer. To diversify the generated ensemble, the max_extra_msa and max_msa_clusters were set to be 16 and 8 respectively (Del Alamo et al., 2022). To sample the conformational landscape more exhaustively, all five AF2 pTM models (checkpoints) were used for inference and the conformations were joined together; For EigenFold, we executed the inference loading the pretrained model weights with default hyperparameter configurations, say $\alpha = 1.0$ and $\beta = 3.0$; Conformation samples from idpGAN were obtained by running their official released code with the default configuration for each evaluation target sequence.

**Potential Energy Evaluation.** In order to evaluate the reweighted ensemble sampled by STR2STR with probability flow, we calculate the potential energy by a force field. Specifically, we perform the energy minimization procedure for each conformation with the side chains packed. After the

protonation (add corresponding hydrogens to each heavy atom), the all-atom conformation is minimized by the Amber-ff14SB force fields using the OpenMM package (Eastman et al., 2017) with implicit solvent GBn2. Similar to Jumper et al. (2021), the harmonic restraints are independently applied to each heavy atom (non-hydrogen) that keep the whole conformation similar to the input, with spring constant being 10 kcal/mol·$Å^2$. The maximal step of minimization is set to be unlimited with tolerance 2.39 kcal/mol. After convergence, the minimized potential energy is reduced to be unit-less by dividing the tempering coefficient $k_B T$, which is approximately 0.5918 mol/kcal for energy unit in kcal/mol.

## E    DETAILS OF EVALUATION METRICS

In this section, we elaborate the definition of the evaluation metrics introduced in the experiments.

**Validity**    The Validity will examine each conformation with respect to (1) clash, and (2) breaking bond. Firstly, the non-clash validity (*Val-Clash*) is defined as the ratio of clash-free conformations. It is calculated as the number of conformations that do not contain steric clashes divided by the number of all evaluating examples. Steric clash is determined by whether two contacting atoms is too close to each other. For an example conformational ensemble $\{\mathbf{x}^{(i)}\}_{i=1}^N$, we have: Val-Clash$(\{\mathbf{x}^{(i)}\}_{i=1}^N) = 1.0 - \frac{1}{N}\sum_{i=1}^N \mathbf{1}\{\exists j, k, \text{ s.t. } |\mathbf{x}_{C^\alpha,j}^{(i)} - \mathbf{x}_{C^\alpha,k}^{(i)}| < \delta\}$, where $\mathbf{x}_{C^\alpha,j}^{(i)} \in \mathbb{R}^3$ indicates the $C^\alpha$-coordinate of $j$'s residue in conformation sample $\mathbf{x}^{(i)}$. In practice, we choose $\delta$ according to the van der Waals radius of $C^\alpha$ minus an allowable overlap $\delta_d$, or formally $\delta = 2 \times 1.7 - \delta_d$ (unit : Å). The default value of $\delta_d$ is set to be 0.4, which is a reasonable value when examining protein-protein interactions (Ramachandran et al., 2011). Secondly, the bonding validity (*Val-Bond*) is defined as the ratio of conformations that maintain the distance between adjacent $C^\alpha$ atoms within certain threshold. The rationale behind is that adjacent $C^\alpha$ are "bonded" by the chemical bonds of $C^\alpha - C - N - C^\alpha$ and should not be too distant from each other. Since there is no general rule defining the valid "bonding" distance for $C^\alpha$-$C^\alpha$, we adopt a statistical threshold which is defined by the maximum adjacent $C^\alpha$-$C^\alpha$ distance among the reference MD (*full*) trajectories of the target protein. By definition, for target protein $P$, we have Val-Bond$(\{\mathbf{x}^{(i)}\}_{i=1}^N; P) = 1.0 - \frac{1}{N}\sum_{i=1}^N \mathbf{1}\{\exists j, \text{ s.t. } |\mathbf{x}_{C^\alpha,j}^{(i)} - \mathbf{x}_{C^\alpha,j+1}^{(i)}| > \delta_{bond}^{(P)}\}$, where $\delta_{bond}^{(P)}$ is the $P$-specific threshold defined above, and other symbols are similar to above.

**Fidelity**    The fidelity of a set of conformations is evaluated by measuring the distributional similarity between the reference ensemble and the distribution of generated ensemble, similar to the Fréchet inception distance (FID) (Heusel et al., 2017) for evaluating the synthetic images. We adopted the Jensen-Shannon (JS) divergence due to its symmetric property that has penalty to the model distribution for lack of ground truth coverage and biased coverage. To cater for baseline models, only $C^\alpha$-atoms are consider to calculate the divergence metrics. Inspired by Janson et al. (2023) and Arts et al. (2023), we adopt three important roto-translation invariant features that can loyally reflect the ensemble characteristics: (1) Pairwise distance. Following the setting of Arts et al. (2023), the pairwise distance is computed by enumerating all pairs of atoms with an offset three. To transform the continuous values into distribution, histograms are built with $N_{\text{bin}} = 50$ bins to represent the categorized pairwise distribution over which the JS divergence is calculated. For each channel of histogram, a pseudo-count value of $\epsilon = 10^{-6}$ was used for zero frequencies to slightly smooth the distribution. (2) The slowest two components of the time-lagged independent component analysis (TICA) (Naritomi & Fuchigami, 2011; Pérez-Hernández et al., 2013). We firstly calculate and flatten the pairwise distances for each protein conformation in the ensemble to be evaluated. We then fit the projection of TICA based on the reference full MD trajectories for each target using the Deeptime Library (Hoffmann et al., 2021). The first and second coordinates are selected after applying the TICA dimension reduction to each ensemble. Note that the samples from neural models are biased with respect to Boltzmann ensemble and time-invariant, and we thus project them uniformly to the TICA directions of the reference MD trajectory for visualize purpose. Histograms are built for both components similar to the pairwise distance above. (3) Radius of gyration, which measures the root mean square distance of each atoms relative to the center of mass. Similar histogram treatment is applied same as above.

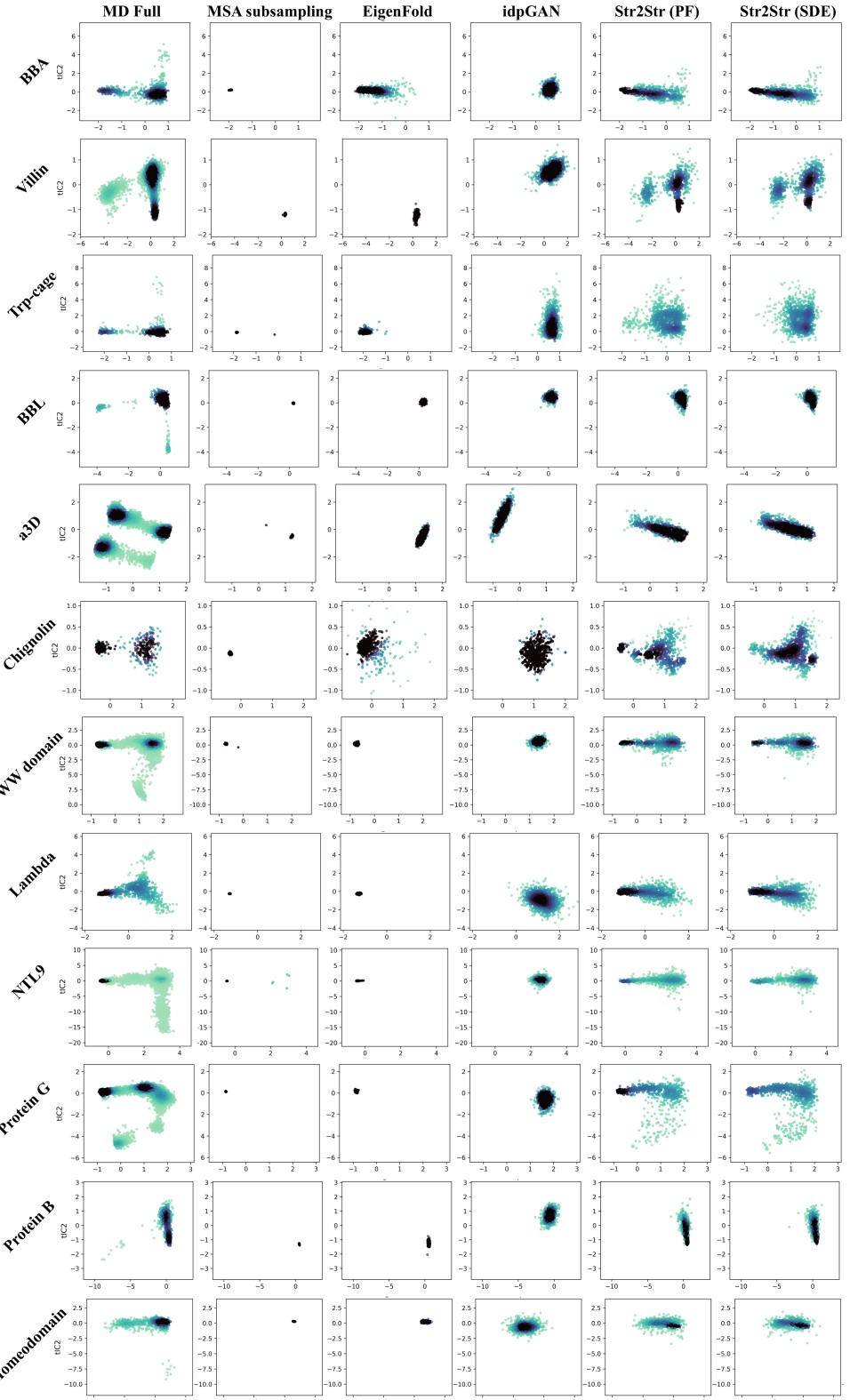

Figure S4: TICA plots for each target protein in the fast folding benchmarks.

**Diversity** The diversity of the ensemble of interest can be derived from any structural similarity score by enumerating and averaging the pairwise scores for that ensemble. Here we adopt two most commonly used scoring functions: root mean square deviation (RMSD) and TM-score (Zhang & Skolnick, 2004). RMSD reflects the deviation degree in length (here we use nanometer (nm) as unit) and is unnormalized. TM-score, on the contrary, is a normalized score to evaluate the structural similarity between two input structures, ranging from 0 to 1 and unit-free. During evaluation, both scores are calculated using the officially released binary from (Zhang & Skolnick, 2004) independently for each fast folding target. After that, we use the mean absolute error (MAE) of both diversity metrics between the reference full MD trajectory and each model and finally report the average across all targets. The rationale is that the ensemble diversity is not the higher the better, which instead depends on the structural characteristics of different proteins. Specifically, some proteins can show rigid structure, of which a typical example is the cyclic peptides (Kessler, 1982). In this case, the motions in the protein dynamics are limited, and thus embody small ensemble diversity. On the other hand, other proteins show more dynamic behaviors like distinguishable different states. They can have relatively larger diversity, which can be loyally reflected in the long MD simulations. For example, the SARS-Cov-2 spike protein was found to have transitions between its open and closed states (Gur et al., 2020). Therefore, we align the ensemble diversity with MD references rather than evaluating as the higher the better.

# F    EXTENDED EXPERIMENTAL RESULTS

## F.1    FAST FOLDING PROTEINS

Here we visualized the contact map and TICA plots for each target from the fast folding proteins, which are shown in Figure S3 and Figure S4, respectively. Following Arts et al. (2023), we use $d = 10\text{Å}$ as the criterion to discriminate contact pair of atoms, and take the logarithm of frequency to color the contact map. STR2STR outperformed other neural sampling baselines by better capturing the metastable states reflected in TICA and contacting dynamics in contact map. By comparing STR2STR and reference MD full, we found that STR2STR basically explored the coarse-grained contact patterns yet the fine-grained characteristics of the sampled structures can be further improved as the TICA plots are not perfectly matched. One limitation of STR2STR, due to learning simply from crystal structure, is that for some cases, it failed to well explore the transient states (colored in shallow green). These points, with low probability or high free energy, are explored by MD simulations via climbing the energy barriers and do not present during the sampling of STR2STR.

## F.2    CONFORMATION INPAINTING OF NANOBODY

In this study, we investigated the inpainting capability of our model on a nanobody derived from Llama glama (PDB entry: 1G9E). The dynamic nature of CDR loops in nanobodies underpins their binding capacity to specific targets. To sample the conformation CDR loops, we initialized from one of the PDB structures, froze the nanobody frame and only perform STR2STR translations for the three CDR loops according to with IMGT numbering (Lefranc et al., 2003). To enhance the loop modeling, the sampling is based on the $r_c = 1.0$ training checkpoint and SDE with other configurations setting as default. As illustrated in S5, the loop structures produced by STR2STR are comparable to the native configurations seen in NMR structures, which demonstrates the potential of STR2STR to perform conformation inpainting.

## F.3    VISUALIZE CONFORMATIONS OF VARYING $T_\delta$

In this section, we showcase by visualizing the protein conformations from different $T_\delta$ in Figure S6. To enhance the distinction, we color the protein ribbons according to their secondary structure assignment: helix is colored red, beta sheet is colored blue and coil (loop) is colored brown. As shown in Figure S6, increased $T_\delta$ renders sampled conformations to possess distinguishable differences while keeping certain characteristics of the initial structure. Additionally, we show the moving TM-diversity of the sampled ensembles along the increasing $T_\delta$ in Figure S7, from which we find that the overall diversity plateau after around $T_\delta = 0.8$ for both PF and SDE.

To better illustrate the effect of $T_\delta$ on sampling, we compare the sampled conformations of WW domain between STR2STR and the reference MD simulation in the two-dimension TICA space. As

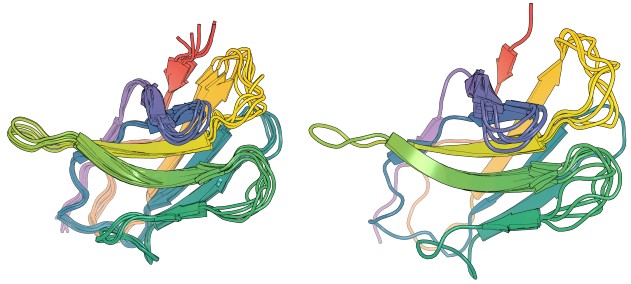

Figure S5: Visualization of nanobody conformation inpainting by STR2STR (SDE). On the left, the structure ensemble is derived from the NMR structures (PDB entry: 1G9E). On the right, we present the inpainted structures of CDR loops 1 to 3.

Table S8: Pearson correlations between sampled diversity and apo-holo diversity, measured in TM-score or residue flexibility as in Jing et al. (2023). Results of EigenFold is obtained from the original paper. The mean/median correlations are reported for per-target residue flexibility.

| Method | TM | Residue flexibility (global) | Residue flexibility (per-target) |
|---|---|---|---|
| EigenFold | 0.12 | 0.13 | **0.41/0.4** |
| STR2STR (PF) | **0.21** | **0.18** | 0.33/0.31 |
| STR2STR(SDE) | 0.11 | 0.13 | 0.33/0.34 |

shown in Figure S8, we plot the scatter of 100 samples from STR2STR in both $T_\delta = 0.3$ (relatively small perturbation) and $T_\delta = 0.7$ (relatively large perturbation), in comparison with that of reference MD. It demonstrates that STR2STR can sample the distant mode and exhibit some unfolded patterns. However, one can still differentiate the fine-grained structural deviation which is not reflected in the low-dimensional TICA space, which may explain the performance gap between JS-PwD and JS-TIC in Table 1.

### F.4   APO/HOLO DIVERSITY

In this section, we evaluate the performance STR2STR on the Apo/holo dataset (Jing et al., 2023), which contains 90 pairs of apo/holo records in PDB. Following Jing et al. (2023), we sample 5 structures per apo/holo pair using STR2STR and evaluate them using the same script as in EigenFold. As shown in Table S8, STR2STR with probability flow (PF) shows better correlation with the ground truth diversity, yet worse on the per-target correlations. The performance inconsistency between EigenFold and STR2STR is due to that the per-target correlation can be affected by the varying residue lengths.

Moreover, following Jing et al. (2023), we scatterplot the TM-scores of each apo/holo target for both EigenFold and STR2STR. In specific, we investigate (i) the ability to capture both apo/holo using the *ensemble TM-score* ($\text{TM}_{\text{ens}}$) and (ii) the diversity of the sampled conformations with the *diversity TM-score* ($\text{TM}_{\text{var}}$). As shown in Figure S9, the EigenFold generates better apo/holo structures with overall higher ensemble TM-scores ($\text{TM}_{\text{ens}}$). This merit can come from the OmegaFold embeddings on which the EigenFold is conditioned, while the forward-backward process of STR2STR alone is not sufficient to recover accurate fine-grained structures. On the other hand, from the lower plots, we found that STR2STR (PF) has a apparently better correlation (0.21 v.s. 0.12, in Table S8) between predicted diversity $\text{TM}_{\text{var}}$ and ground-truth diversity $\text{TM}_{\text{apo/holo}}$, which indicates the STR2STR aligns better with the underlying dynamic characteristics. Inspired by this finding, one promising future work can focus on jointly modeling accurate fine-grained structures like EigenFold/AF2, while capturing diversity as well as STR2STR.

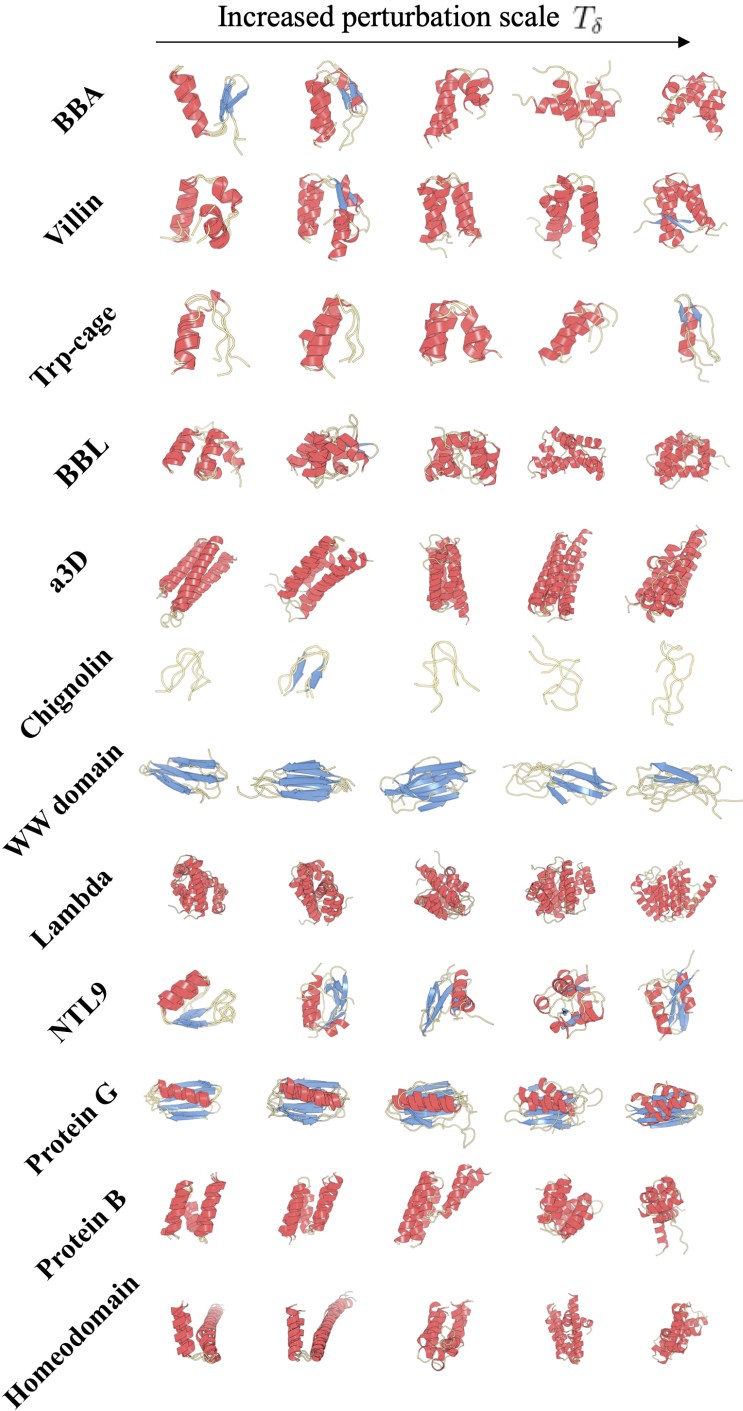

Figure S6: Visualize of protein conformation sampled by STR2STR under different $T_\delta$ (from left to right: 0.3, 0.4, 0.5, 0.6, 0.7) for the fast folding targets. Helical residues are colored red, beta sheets are colored blue and coils are colored brown.

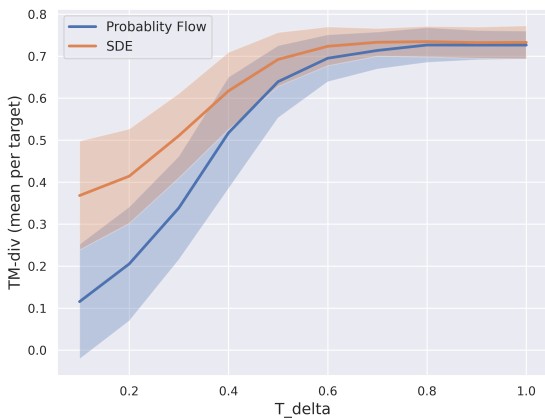

Figure S7: Moving TM-diversity of the sampled ensemble along increasing $T_\delta$. The TM-diversity at each $T_\delta$ is averaged across all per-target TM-diversity for STR2STR in PF or SDE settings.

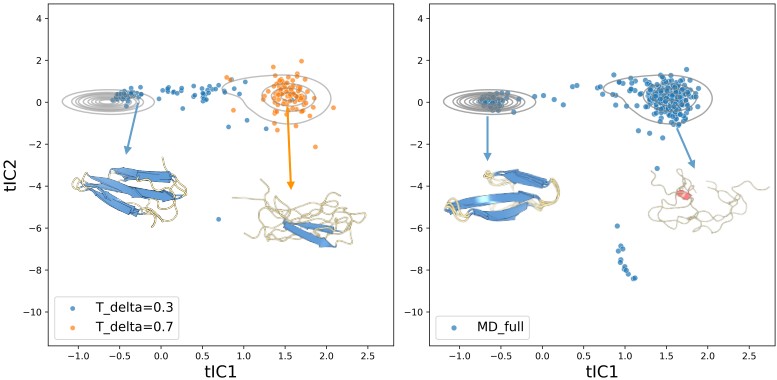

Figure S8: The illustrative diagram for sampling WW domain conformations. (left) STR2STR under different perturbation $T_\delta$, being compared with (right) long MD simulations. The grey contour is the kernel density estimate (KDE) plot based on the whole reference MD trajectories.

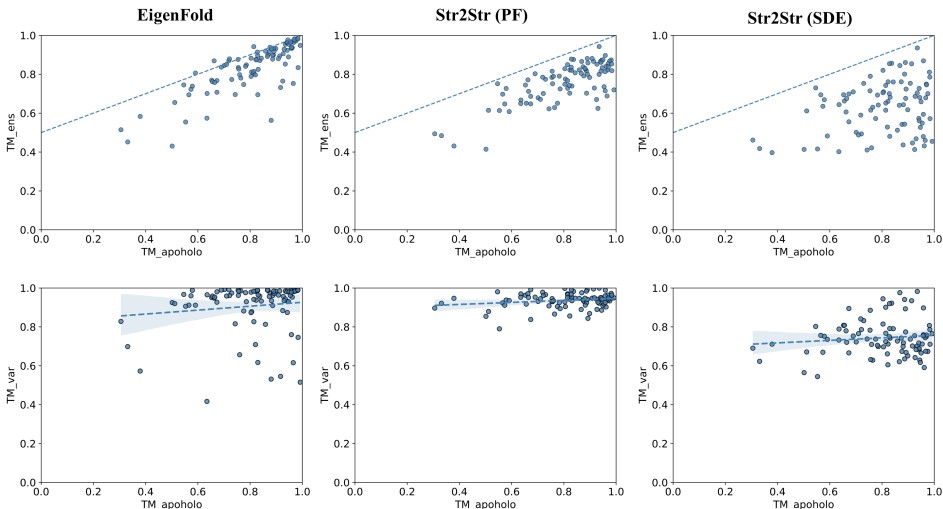

Figure S9: The scatterplot of the TM-score on (upper) $\text{TM}_{\text{ens}}$ versus $\text{TM}_{\text{apo/holo}}$, and (lower) $\text{TM}_{\text{var}}$ versus $\text{TM}_{\text{apo/holo}}$ for EigenFold and STR2STR, following Jing et al. (2023). Note that for the lower plots, the regression line is shown with confidence interval. We found that EigenFold can generate structures with higher quality and thus achieve better ensemble TM-score ($\text{TM}_{\text{ens}}$), by conditioning on the OmegaFold embedding; while STR2STR has better predicted diversity ($\text{TM}_{\text{var}}$) correlation with the ground truth diversity.

## F.5   TILTING TOWARDS BOLTZMANN DISTRIBUTION

The data-driven diffusion sampler STR2STR may not guarantee the generated samples are from the Boltzmann distribution, but instead, an ensemble of independently sampled conformations. To tilt the generated samples to obey this physical assumption, we can compute the likelihood of samples from probability flows with the learned score. By approximately replacing the conformation log-likelihood $\log p_X(\mathbf{x})$ by the amortized translation distribution $p_\theta$ over frames with learned parameters $\boldsymbol{\theta}$:

$$\log p_X(\mathbf{x}) \approx \Delta V(\mathbf{T}) + \log p_{\boldsymbol{\theta}}(\mathbf{T}|\mathbf{T}_0) + \log p(\mathbf{T}_0)$$

$$= \Delta V(\mathbf{T}) + \log p_{T_\delta|0}(\mathbf{T}_{T_\delta}|\mathbf{T}_0) - \frac{1}{2} \int_0^{T_\delta} g^2(t) \nabla \cdot \mathbf{s}_\theta(\mathbf{T}_t, t) dt + C, \qquad (17)$$

where $\Delta V(\mathbf{T}) := -\log[\det[(J_{\Gamma_{\text{bb}}}(\mathbf{T}))]]$ is the change of volume for reconstructing Euclidean coordinate from frames and $J_{\Gamma_{\text{bb}}}$ is the corresponding Jacobian matrix; $\nabla\cdot$ is the divergence operator, with the prior likelihood $\log p(\mathbf{T}_0) \equiv C$ being constant for a fixed ensemble. We can then reweight the conformation ensemble by calculating the importance weights:

$$w(\mathbf{x}) \propto \exp\left[-\left(u(\mathbf{x}) + \log p_X(\mathbf{x})\right)\right], \qquad (18)$$

where $u(\cdot) : \mathbb{R}^{3N} \to \mathbb{R}$ is some all-atom force field for proteins with its unit reduced by $1/k_B T$, where $k_B$ is the Boltzmann constant and $T$ the simulation temperature. The generation with reweighting can approximately achieve unbiased sampling from Boltzmann distribution (Noé et al., 2019).

Table S9: Distributional divergences on fast folding proteins (Lindorff-Larsen et al., 2011) for probability flow STR2STR w/ and w/o reweighting.

| Setting | JS-PwD | JS-TIC | JS-Rg |
|---|---|---|---|
| Str2Str (PF) | 0.375 | 0.397 | 0.448 |
| Str2Str (PF, reweighted) | 0.487 | 0.461 | 0.489 |

However, in practice, we found that the reweighting for zero-shot conformation sampling using STR2STR cannot improve the distributional metrics as shown in Table S9. The reason may lie in

the fact that, unlike previous Boltzmann generators (Noé et al., 2019; Jing et al., 2022), STR2STR generates conformation hypothesis by repurposing the translation distribution $p_{\boldsymbol{\theta}}(\mathbf{T}|\mathbf{T}_0)$ in a transfer learning manner, and receives no energy supervision during training of STR2STR to induce close matching to equilibrium distribution. Moreover, the all-atom force field can be sensitive to the position of side-chain atoms, which are however predicted by an imperfect external module, and thus make the estimator in Eq. (18) suffer from large variance. Still, the fidelity metrics (JS-*) used in the benchmark can also reflected (penalized) the the potential bias between the model distribution and underlying Boltzmann distribution represented by MD trajectories. The improvement of STR2STR to accommodate energy-based training for score-based STR2STR is a promising future work.

## G    EXTENDED DISCUSSION

To better contextualize the proposed STR2STR, we review a much broader range of related and relevant literature as follows. In Section G.1, the traditional molecular dynamics and Monte Carlo methods are overviewed. In Section G.2, we discussed in detail the STR2STR with related methods that predicting or generating the protein structures.

### G.1    MOLECULAR DYNAMICS AND MONTE CARLO METHODS

In this section, we overview the related works of Molecular dynamics (MD) and Monte Carlo (MC) methods for protein conformation sampling. As described in previous sections, MD evolves the Newtonian equation along time explore the conformation space guided by some force field, such as Amber (Wang et al., 2004) and CHARMM (Vanommeslaeghe et al., 2010). Although MD is simple and considered effective, the simulation can be computational intractable to study natural dynamic behaviors such as folding (Lindorff-Larsen et al., 2011). To accelerate, several strategies sacrifice accuracy in trade for speed, such as using implicit solvent (Ferrara et al., 2002) or coarse-graining (de Jong et al., 2013). Enhanced sampling methods have been proposed to enforce the exploration of MD simulations (Abrams & Bussi, 2013) and make long timescale accessible with mild computation. To name a few, the umbrella sampling (Torrie & Valleau, 1977), metadynamics (Laio & Parrinello, 2002) and replica exchange molecular dynamics (REMD) (Hansmann, 1997; Sugita & Okamoto, 1999; Swendsen & Wang, 1986). However, none of these methods can totally replace canonical MD by providing completely comparable thermodynamics and kinetic characteristics.

MC simulation typically steers a Markov chain Monte Carlo (MCMC) trajectory to explore the protein conformation space (Fichthorn & Weinberg, 1991). Due to lack of temporal information, MC methods can only yield samples thermodynamic ensemble and cannot reveal insights into kinetics directly (Paquet et al., 2015). Nevertheless in Liang & Wong (2001), the authors showed that large-scale kinetics can be reconstructed from the MC-sampled ensembles. Since the MC moves are not driven by the gradient (force) field, typically only a few degrees of freedom can be altered in a single move, which can be hindered from using explicit solvents (Nerenberg & Head-Gordon, 2018) and scaling up to large systems. In Heilmann et al. (2020), the authors showcased the MC simulations on 3 (Trp-cage, Villin and WW domain) out of 12 fast folding proteins in comparison with the MD counterpart (Lindorff-Larsen et al., 2011). But the result is less convincing than MD since the folding temperatures are overestimated. A most recent work (Klein et al., 2023) leverage the normalizing flow to build a neural MC sampler by training on MD simulation data. However, such model is only built for very short peptides (2-4 amino acids) and the generalization capacity is still questionable. Currently, MC falls out of the mainstream for studying thermodynamics and kinetics for macro-molecular systems such as protein.

### G.2    PREDICTION AND GENERATION FOR PROTEIN GEOMETRY

In this paper, we have proposed the STR2STR for zero-shot conformational sampling which leverages the bidirectional diffusion dynamics in SGMs. This method shares many aspects with structure prediction and generation methods. Structure prediction models such as Jumper et al. (2021); Baek et al. (2021); Lin et al. (2023) aim to recover the ground protein folding state by learning on crystal structures in PDB and exhibit highly accurate accuracy (Jumper et al., 2021). The EigenFold (Jing et al., 2023) formulated the structure prediction task as conditional generative modeling, and assumes there exist multiple stable conformations (conformers) for a single amino-acid sequence.

However, one of the biggest problem is the lack of ground truth multi-state structure data that belongs to a single sequence for training the conditional diffusion model. Methods for protein backbone design, as described in Section 5, aim to learn the unconditional distribution of PDB structures and generate *de novo* backbone or scaffold structures. These backbone generative models (Ingraham et al., 2022; Trippe et al., 2022; Watson et al., 2022; Wu et al., 2022a; Yim et al., 2023) model the protein structure landscape and successfully yield protein-like decoys with good "designability". None of these methods discover the potential of such models for conformation sampling. Several recent works (Del Alamo et al., 2022; Vani et al., 2023; Barrio-Hernandez et al., 2023) adopted a modified MSA input of AlphaFold2 to diversify the predicted samples but this protocol still exhibits limited structural deviation since the training objective of AF2 is deterministic.

Our proposed STR2STR can have closed relation to and well based on above research directions. Firstly, to transfer the modeled structure distribution to a sampling proposal, STR2STR relies on a folding module such as ESMFold (Lin et al., 2023) that provides a starting point. Secondly, both EigenFold and STR2STR leverage the PDB database for training with a denoising score matching (DSM) objective and aim to sample multiple structures. The difference is that, the EigenFold modeled a mapping from the sequence-encoded latent space (say OmegaFold (Wu et al., 2022b) embeddings $(s_i, z_{ij})$), to the structure space. In contrast, the STR2STR delineated solely the structure space and sampled conformations in an amortized manner. The performance gap between them can be because Eigenfold handles a more challenging conditional generative modeling task, where the embedding condition $(s_i, z_{ij})$ is far complicated than a typical scalar class label. Without enough data for each condition in PDB, the resulting diversity (exploration) is thus greatly limited. In contrast, STR2STR learns simply the structure space where the conformations can be representation as data-like points, and thus perform better. Lastly, the amortized learning objective in Eq. (8) coincides with the one used in the (unconditional) generative modeling of backbone structures. Intuitively, one can view the STR2STR as an inference-time **plug-and-play** of backbone generative models by applying the forward-backward dynamics. Our results demonstrate that learning on purely structure space can be effectively transferred to specific conformation sampling.

The core assumption behind the proposed framework STR2STR is that the dynamic patterns among conformations can be partially shared with the patterns between different structures in the PDB. That is to say, the common **inter-protein** structural characteristics in the training set can be distilled by DSM in the score network, which is "unlocked" during sampling to infer the **inter-conformation** deviation within the test system. Our work opens up a broad research direction that leverages the generative modeling on structure alone to tackle the protein conformation sampling task.

