# OpenReview forum: "Str2Str: A Score-based Framework for Zero-shot Protein Conformation Sampling"
_ICLR.cc/2024/Conference — ICLR 2024 poster_

### Official Review · Reviewer_6bVL · 2023-10-28

**Soundness:** 3 good
**Presentation:** 3 good
**Contribution:** 3 good
**Rating:** 6
**Confidence:** 3

**Summary:**

This paper aims to capture the dynamic protein structures and obtain various protein conformations. Instead of the resource-demanding MD and MC methods, the authors take inspiration from simulated annealing and propose a structure-to-structure translation framework for zero-shot conformation sampling. Roto-translation equivariance is appropriately guaranteed. Experiments on the 12 proteins in a newly established benchmark confirm the validity, fidelity, and diversity of the produced conformations. Ablation analysis and case study are provided for readers to have a deeper understanding of this paper. The problem under study is significant and this proposed idea is interesting.

**Strengths:**

- Strucutre-to-structure translation seems a novel way to sample various conformations of a target protein. It requires less computational resources than MD and MC.
- The experiments are described in details, which is very helpful for the readers the understand the proposed method.
- The paper is well-written and easy to follow. More explanation about the formulas would further increase readability.

**Weaknesses:**

- Although the overall structure-to-structure translation for protein conformation sampling is novel, there may not be adequate points to support the whole paper. The forward-backward process is most similar to a diffusion process, where the forward process adds noise (or heating) to data distribution and the backward process denoise (or annealing). The learning objective is similar to DenoisingIPA, except the minor edge translation layer.
- Proper discussion of MC and MD-based related studies for protein conformation sampling could give readers a more thorough picture of where this paper is located.
- Also, I'd like to see MC and MD-based baselines in the experiments. Though they are slow, I'm wondering how they perform in terms of validity, fidelity and diversity.

**Questions:**

- What is the total number (or dimension) of variables T, R, v, X? Does this have something to do with the number of residues in each protein?
- The score matching objective is the supervision signal. That is, the model is required to approximate the distribution of atom positions/angles of a target protein. I'm wondering how the proposed method encourages diversity of the sampled conformations.
- Please properly use the term dynamics. Not sure if the authors properly interpret this term. To the best of my understanding, the concept of dynamics should have something to do with time and is usually used to indicate a temporally changing variable. Here in this paper, I do not see the conformation of a target protein changes with time. Instead, this paper cares more about different candidates of stable conformations. The proposed forward-backward process seems like a dynamics but essentially is a process of heating-then-simulated-annealing, which is different from the dynamics of protein conformation. Please properly modify some statements, especially those regarding dynamics.
- How similar are the 12 proteins in the benchmark to the training protein data? Need to ensure zero-shot by quantifying the dissimilarity of train/test proteins at sequence and structure levels.

---

> ### Author Response · Authors · 2023-11-18
> **Response to Reviewer 6bVL: Part 1**
>
> Firstly, we sincerely thank the reviewer for providing helpful suggestions to improve our manuscripts. Specifically, we want to address the above concerns as follows:
>
> **Weakness:**
>
> **Seemingly “incremental” to previous methods.**
> - Thanks for raising this point. We want to humbly argue in the following aspects:
>   - Firstly, the focus of this research paper is to present a zero-shot framework for the task of conformation sampling w/o expensive simulation data. As far as we are concerned, Str2str is the first-in-class example that can sample protein-level conformations in the zero-shot manner.
>   - One of our main contributions is the successful experimental demonstration that **it is able to transfer the generative models pretrained on PDB with score matching objective to the downstream task of conformation sampling**, even in a zero-shot manner. This distinguishes our work from those directly applying well-known methods such as diffusion to new data in an old task framework, and thus should not be criticized.
>   - Lastly, we are not aiming at proposing new generative methods extending SGM/diffusion [1] or introducing a new structure decoder on top of IPA as in AlphaFold2 [2]. Correspondingly, beyond Diffusion and IPA, our framework can flexibly accommodate other counterparts, such as LDM[3], flow matching[4] or SE(3)-transformers[5], etc, and these can be future work.
>
>
> **More related works for MC/MD methods.**
> - Thanks for this valuable suggestion. To better contextualize, we have carefully reviewed the related MC/MD literatures and complemented the relevant discussion in the Appendix G.1.
>
> **Baselines of MC and MD.**
> - Firstly, we note that the MD baseline, obtained from D.E. Shaw Research [4], were already shown in our main results. These simulations are among the most authentic and well-recognized studies in the MD domain, and unique in the simulation time scale.
> - Secondly, for MC, their validity should be comparable to MD since both of them involved force fields. Since MC is not mainstream in the MD domain (since it is not scaling well), there are very few works studying the protein dynamics using MC in recent decades.
>   - Evaluating MC methods on fidelity/diversity requires sampled trajectories, which are not accessible. Running any MC or MD in the same time scale as [5] from scratch is computationally intractable for us. The estimated simulation time for all benchmark systems far exceeded >10,000 GPU days, which is comparable to LLM pre-training such as Llama[7]. The most relevant work [8] conducted a MC-based study using a 100-fold larger timestep (x100 acceleration), it still requires ~100 GPU days and their data is not open-sourced.
>
> References:
>
> [1] Song, Yang, et al. "Score-Based Generative Modeling through Stochastic Differential Equations." International Conference on Learning Representations. 2020.
>
> [2] Jumper, John, et al. "Highly accurate protein structure prediction with AlphaFold." Nature 596.7873 (2021): 583-589.
>
> [3] Rombach, Robin, et al. "High-resolution image synthesis with latent diffusion models." Proceedings of the IEEE/CVF conference on computer vision and pattern recognition. 2022.
>
> [4] Lipman, Yaron, et al. "Flow matching for generative modeling." arXiv preprint arXiv:2210.02747 (2022).
>
> [5] Fuchs, Fabian, et al. "Se (3)-transformers: 3d roto-translation equivariant attention networks." Advances in neural information processing systems 33 (2020): 1970-1981.
>
> [6] Lindorff-Larsen, Kresten, et al. "How fast-folding proteins fold." Science 334.6055 (2011): 517-520.
>
> [7] Touvron, Hugo, et al. "Llama: Open and efficient foundation language models." arXiv preprint arXiv:2302.13971 (2023).
>
> [8] Heilmann, Nana, et al. "Sampling of the conformational landscape of small proteins with Monte Carlo methods." Scientific reports 10.1 (2020): 18211.

---

> ### Author Response · Authors · 2023-11-18
> **Response to Reviewer 6bVL: Part 2**
>
> **Questions:**
>
> **Shape of the mentioned variables.**
> - They are dependent on the number of residues: the shape of respective variables for a single protein are: T = (N_residue, ); R = (N_residue, 3, 3); v = (N_residue, 3); x = (N_residue, 37, 3).
>
> **Explanation of the sampling diversity.**
> - Thanks for asking this question. We want to clarify that the proposed Str2str is in a transfer learning setting which does not require distribution data of the test system for training. During sampling, the perturbation (forward) process enforces exploration (say diversity) and the amortized learned score network will anneal (exploit) the perturbed sample to a candidate conformation. Overall, the forward-backward process achieves good balance between exploration and exploitation.
>
> **Explanation of the term “dynamics” in Section 3.2.**
> - Explanation: The semantic meaning of “dynamics” used in the forward-backward dynamics originates from the Langevin dynamics — which the backward process of diffusion/SGMs can belong to. We agree that “dynamics” comes with a time, while the time indicates here the time index of the diffusion process. According to your suggestion, we have altered the writing by using “process” in replace of “dynamics” in the Section 3.2. We hope this could address your concern.
>
> **Whether to ensure dissimilarity between train-test sets.**
> - Thanks for asking this question. In case there may be the least misunderstanding, we want to humbly clarify that such a process is not necessary: (1) amino-acid sequence information is neither used as input nor output; (2) The resulting evaluation is conducted based on the distribution metrics instead of the accuracy for a single structure. In other words, the setting is unsupervised w.r.t. the conformation sampling task; (3) The sampling for each test system corresponds to a new task during inference, and not a single data point belongs to that task is present during training. It is by definition zero-shot learning; (4) From the ML point of view, the usual train-test dissimilarity is enforced to avoid data leakage from supervision, which can make the “generalization” non-sense. This does not happen in our case.
> - Therefore, we preprocessed the training data by excluding the PDB entries to contain no a single conformation of all test systems.
>
>
>
> We sincerely hope our response can address your concerns. If you have other questions, we are very glad to discuss them during the rebuttal.

---

> ### Author Response · Authors · 2023-11-21
> **Look forward to further feedbacks during the reviewer-author discussion period**
>
> Dear Reviewer 6bVL,
>
> Thank you for your helpful suggestions for improving our manuscripts. We sincerely appreciate your feedback and have carefully responded to each question point-by-point.
>
> This is a kind reminder that as the reviewer-author discussion period is ending soon, we look forward to hearing from you about your feedback to our response. (Unlike previous years, there will be no second stage of reviewer-author discussion this year)
>
> In particular, we have complemented related discussions and improved the clarification of writing, according to your advice.
>
> Thank you again for your precious time and effort!
>
> Best regards,
>
> Authors #3973

---

### Official Review · Reviewer_Fe1f · 2023-10-30

**Soundness:** 3 good
**Presentation:** 3 good
**Contribution:** 3 good
**Rating:** 6
**Confidence:** 4

**Summary:**

The authors present Str2Str, a score model for protein conformation sampling. The model is trained on crystal structures from the PDB and could generate diverse conformations for unseen protein systems. Unlike previous mothods (e.g., EigenFold), Str2Str is sequence-agnostic. The model learns protein conformation distribution without knowing the amino acid residue sequence. Through the proposed forward-backward dynamics, which is similar in vein to simulated annealing, the model is able to effectively explore different potential energy minima without suffering from rare event sampling problems.

I find this manuscript well-written and provides promising results towards solving the protein conformation sampling challenge. Please see some of my main concerns below.

**Strengths:**

- The major contribution of this work is the proposed forward-backward dynamics using a structure-based score model trained on the PDB dataset. From Fig. 5 and Fig. S3, it is clear that Str2Str outperforms other existing protein conformation sampling methods.

**Weaknesses:**

See questions.

**Questions:**

- What is the difference between the proposed Str2Str pipeline and applying forward-backward dynamics using a pretrained FrameDiff?

- Is it possible to provide an additional ablation study on a few fast-folding proteins, where only alpha-Carbon coordinates are modeled? Maybe you can just turn off the rotation loss. I am curious whether frames are essential for accurate protein conformation sampling.

- From Appendix >> Inference Stage on page 22, the ensemble structures is obtained by merging samples from each perturbation scale, in total 1,000 conformations for fast-folding proteins. It is surprising that metrics such as JS-PwD/MAE-TM remains so small even though $T_{\delta}$ can be as large as 0.7. I expect that many samples are quite different from the starting conformation, especially for large $T_{\delta}$. Could you please explain this behavior?

- Since the model is sequence-agnostic, the model should sample many "outlier" conformations, which is not accessible from the given protein sequence. In Fig. S5, are most of the shown structures valid conformations for each protein system?

- Page 18, I do not understand why reweighting could not help improve relevant distributional metrics. Does that mean the model likelihood estimation is not accurate, or model diversity originates from model uncertainty? Quick check, if we manually make $\log p_{X}$ a constant value across all samples, would it improve the metrics?

- Would you mind showing model performance on the apo/holo dataset for a fair comparison with EigenFold?

- Could you please (1) provide the source code in supplementary materials; (2) make the benchmark data open-access for reproducibility?

---

> ### Author Response · Authors · 2023-11-18
> **Response to Reviewer Fe1f: Part 1**
>
> We appreciate the insightful questions and helpful suggestions for improving our manuscript. We have appended the related experiments in the updated PDF. Our response to each question is as follows:
>
> **Questions:**
>
> **Difference between Str2str and Framediff + Forward-backward (FB) dynamics.**
> - Firstly, according to our experimental results in the paper, the amortized conformation sampling can be competent by a backbone generative model trained on PDB. Intuitively, our framework is ready to accommodate all backbone design methods [1,2,3,4] including FrameDiff [4] in a plug-and-play manner, by simply applying the FB dynamics. Among them, FrameDiff that consists of the Riemannian diffusion on $SE(3)^n$ and the IPA module of AF2[5] was acknowledged by us as a very effective score model to follow.
> - Secondly, we argue that Str2str is targeting a totally different task, based on an existing model Framediff: Framediff (and its counterpart [1,2,3]) aims to unconditionally sample novel and designable backbones; Str2str aims to approximate the underlying conformation distribution similar to MD. The model architecture coincidence comes from (as our main contributions) how we designed the zero-shot sampling pipeline and demonstrated the effectiveness of transfer learning and thus should not be criticized.
> - Specific implementation differences: (i) the ideal frame geometry (idealized bond lengths and angles) used during training and inference is sequence-specific instead of all alanine;  (2) the training data is set to be in length between 10 and 512. (that of framediff is >60, which is larger than most test systems), etc.
>
> **Ablation on orientation (rotation matrix) modeling.**
> - The original purpose of using “frames” is to be able to generate all heavy atom coordinates plus the packing module, where the orientation can be important. According to this suggestion, we newly conducted the training w/o rotation and compared it with the previous checkpoint trained w/ rotation. Given the limited time of rebuttal, both checkpoints are aligned to be 300,000 steps for fair comparison. As shown in Appendix C.4, modeling the orientations can be important for the zero-shot conformation sampling.
>
> **Explain the difference between high $T_\delta$ sample and initial sample.**
> - An increased perturbation scale encourages structural diversity and the samples can be different from the starting conformation. Such phenomenon does not contradict with the resulting small JS-PwD/MAE-TM, because these metrics are compared with long reference MD: JS = JS(p_pred||p_MD); MAE = |diversity(pred) - diversity(MD) |. After sufficiently long simulation, MD trajectory also discovers the conformation states that are quite distant from the starting one. The “small” resulting metrics demonstrate the model can mainly agree with the MD with small discrepancy (JS, MAE).
>
>
> References:
>
> [1] Watson, Joseph L., et al. "De novo design of protein structure and function with RFdiffusion." Nature 620.7976 (2023): 1089-1100.
>
> [2] Ingraham, John, et al. "Illuminating protein space with a programmable generative model." BioRxiv (2022): 2022-12.
>
> [3] Lin, Yeqing, and Mohammed AlQuraishi. "Generating novel, designable, and diverse protein structures by equivariantly diffusing oriented residue clouds." arXiv preprint arXiv:2301.12485 (2023).
>
> [4] Yim, Jason, et al. "SE (3) diffusion model with application to protein backbone generation." arXiv preprint arXiv:2302.02277 (2023).
>
> [5] Jumper, John, et al. "Highly accurate protein structure prediction with AlphaFold." Nature 596.7873 (2021): 583-589.

---

> ### Author Response · Authors · 2023-11-18
> **Response to Reviewer Fe1f: Part 2**
>
> **Questions (continued):**
>
> **Discussion on the possible sampling of outlier.**
> - Since there is no criterion defining “valid” conformation, let us denote “valid” as local minima belonging to the underlying distribution. For fast folding proteins, we can infer from the results in [6] that distant (if compared with a folded start) yet valid conformations are transient unfolded states. In Figure S6, large $T_\delta$ tends to destroy the original fold, while keeping some local secondary structures. Especially in the case of WW domain (we complement a Figure S8), the main components become coil (loop) and loosened for large $T_\delta$ .
> - On the other hand, using three JS divergences instead of simply comparing the diversity takes the outlier into consideration during evaluation. The “diversity” roughly reflects the radial of the space coverage while the “JS” penalizes the mistakes by sampling outliers in the wrong direction. The current gap between Str2str and MD reference represents one can still improve this imperfection in the future. In our humble view, even conditioning on sequence can neither guarantee the sampling of “valid” conformation, if without enough training data.
> - Lastly, as a typical “invalid” or “outlier” example, the idpGAN [7] has a strong tendency to sample coil components, which can belong to the “outlier” for most structural proteins. Therefore, the used fidelity metrics differentiate and penalize them based on the MD reference.
>
> **Investigation of reweighting (importance sampling).**
> - According to your suggestion, we have re-evaluated the ensemble w/ logp(X)=Constant (i.e. only use the FF to reweight). The JS-divergence results are [JS-PwD=0.471; JS-TIC=0.448; JS-Rg=0.487], which is still worse than that w/o reweighting. We dived into the details and have the following preliminary analysis:
>   - We noticed the minimized energy for the generated ensembles has much larger standard deviations (~100 kcal/mol) than that in normal MD simulation (<10 kcal/mol) after equilibration. Since the energy is mainly contributed by the side chains, we probe our packing module (FASPR) by re-packing a batch of equilibrated conformation. The minimized energy of the repacked conformations is 50-100 kcal/mol higher than the pre-packed energy. Such results show that, even given perfect backbones, it still fails to generate low-energy side-chain conformation within the minimization steps. Therefore, the importance weight is not stable or accurate, which causes large variance to the estimation. As illustration, energy difference in 5 kcal/mol under 300K will lead to weight $w \approx 4414$. The distribution will be sharply biased toward very few samples, making the estimation much worse by using only 1000 samples. In practice, a more robust packing module can be used to perform effective energy reweighting.
>
> **Apo/holo dataset.**
> - According to your suggestion, we have added the apo/holo benchmark in comparison with EigenFold by strictly following their settings, which is updated in Appendix F.4.
>
> **Code / reproducibility.**
> - We are currently refactoring the codebase into a more user-friendly version. As stated in the Reproducibility Statement paragraph, we promised all source codes, scripts, training data, necessary artifacts and experimental environments (settings) will be released upon the acceptance of this work. The benchmark MD trajectories are obtained from the original authors in D.E. Shaw Research (DESRES, https://www.deshawresearch.com/index.html). According to the DESRES License 3(e), “Licensee shall not…disclose, without DESRES's prior written approval, the Datasets, whether in whole or in part, other than as expressly authorized hereunder…”. It is thus illegal to put them on OpenReview during the double-blind review. For both reproducibility and legal purposes, we will  sincerely ask DESRES whether it is proper/how to share the benchmark data upon acceptance.
>
> References:
>
> [6] Lindorff-Larsen, Kresten, et al. "How fast-folding proteins fold." Science 334.6055 (2011): 517-520.
>
> [7] Janson, Giacomo, et al. "Direct generation of protein conformational ensembles via machine learning." Nature Communications 14.1 (2023): 774.
>
>
> We sincerely hope our response can address your concerns. If you have other questions, we are very glad to discuss them during the rebuttal.

---

> ### Author Response · Authors · 2023-11-21
> **Look forward to further feedbacks during the reviewer-author discussion period**
>
> Dear Reviewer Fe1f,
>
> Thank you for your helpful suggestions and the recognition of the contribution of our paper. We sincerely appreciate your feedback and have carefully responded to each question point-by-point.
>
> This is a kind reminder that as the reviewer-author discussion period is ending soon, we look forward to hearing from you about your feedback to our response. (Unlike previous years, there will be no second stage of reviewer-author discussion this year)
>
> In particular, we have added more experimental results and related discussion per your advice. If you need any clarification, please feel free to contact us and we are very glad to discuss with you.
>
> Thank you again for your precious time and effort !
>
> Best regards,
>
> Authors #3973

---

> ### Comment · Reviewer_Fe1f · 2023-11-22
> **Response to author rebuttal**
>
> Dear authors,
>
> Thank you for your response, which addressed most of my concerns. My evaluation has been updated accordingly.
> Lastly, would you mind showing the TM$_\mathrm{ens}$ plot (EigenFold Fig 5) in the appendix?

---

> > ### Author Response · Authors · 2023-11-22
> > **Response to reviewer Fe1f**
> >
> > Dear reviewer Fe1f,
> >
> > Thank you for recognizing our efforts and helpful feedback for improving the manuscript. Per your suggestion, we have accordingly revised the appendix in Section F.4 Figure S9 with a descriptive paragraph in Page 29.
> >
> > Best regards,
> >
> > Authors #3973

---

### Official Review · Reviewer_dLpk · 2023-11-02

**Soundness:** 3 good
**Presentation:** 3 good
**Contribution:** 2 fair
**Rating:** 6
**Confidence:** 3

**Summary:**

The authors propose STR2STR, a structure-to-structure framework to perform conditional conformation sampling of protein structures. The goal of conformation sampling is to sample new stable, energetically favorable structures from an initial protein structure. DIirecting atomistic modeling of protein structure is often intractable due to the size and high degrees of the system. Rather than sampling the entire structure at once, the authors reformulate the task into multiple conditional sampling tasks. The backbone is generated conditioned on a ground truth conformation. The backbone and ground truth conformation are used to then generate the side chains and carbonyl oxygen on the backbone. The authors present an updated implementation of the invariant point attention from AlphaFold and show that their confirmation sampling process is SE(3) equivariant. Compared to recent deep learning methods, STR2STR has comparable validity while improving fidelity and diversity of generated conformations.

**Strengths:**

* The authors present a novel representation decomposition for protein structures to enable structure-to-structure translation using a diffusion model.
* The amortized learning objective which uses only pre-confromed data and does not require new simulated sets is significant and can reduce the cost of training future models.
* The improved diversity and fidelity of the model while not relying on simulated conformations is also significant. Improvements over EigenFold, a similar diffusion model is interesting.

**Weaknesses:**

* The benchmarks set is small and make it difficult to judge the results provided.
* Some presentation issues (refer to clarifications)

**Questions:**

* “We notice that the ensemble diversity is not the higher the better and depends on the characteristics of the target system”
Could you please elaborate more on this?
* Is there a reason run times are provided only for a single target protein rather than all 12 (other than the high cost of sampling)?
* Any particular reason why 100ns MD simulation runtimes are not compared with STR2STR runtimes?
    * According to Table 1, smaller trajectories sometimes outperform the proposed model. Having all the data allows the reader to have a more complete understanding of the capabilities of the proposed model

---

> ### Author Response · Authors · 2023-11-18
> **Response to Reviewer dLpk**
>
> We sincerely thanks for your questions and helpful suggestions for improving our manuscript. We put our response to your concerns accordingly as follows:
>
> **Weakness:**
>
> **The size of the benchmark/test set.**
> - First we understand this concern. In fact, the authentic protein trajectory data (long enough) is very limited in the MD domain. Most previous MD research works did not make their data disclosed. The 12 fast folding protein trajectories run by the DEShaw Research are one of the most well-recognized. Though N=12 seems far fewer than common ML benchmarking practice, it is still one of the largest scales for test systems. For example, one related work [1] only chose a subset (N=5) out of these 12 to evaluate their performance.
>
> **General presentation issue.**
> - Thanks for your suggestions. Several clarification issues suggested by other reviewers are addressed by us. Meanwhile, we will pay attention to the overall writing of this manuscript.
>
>
>
> **Questions:**
>
> **Why is not diversity “the higher the better”?**
> - This observation comes from the structural characteristics of different proteins. (i) Some proteins can show very deterministic structure, an extreme example is the cyclic peptides. In this case, the motions in the protein dynamics are limited, and thus with small ensemble diversity from MD simulations; (ii) On the other hand, other proteins show more dynamic behaviors like distinguishably different states. They can have relatively larger diversity, which can be loyally reflected in the long MD simulations. For example, the SARS-Cov-2 spike protein was found to have transitions between its open and closed states [2]. So it is reasonable to compare with a MD-reference diversity rather than claiming it is “the higher the better”.
>
> **Time profiling is shown in Table 2 for one system only.**
> - The runtime profiling on one target is enough to demonstrate the efficiency of the proposed Str2str as (1) the runtime scale is significantly smaller; (2) different systems can share a similar maginitude of simulation time. To better illustrate, we profiled for 100us MD the smallest system (Chignolin) which takes ~95.2 GPU days, and the largest system (A3D) which takes ~200.8 GPU days.
>
> **The reason to profile 100us MD but not for 100ns.**
> - In MD domain practice, running for a scale between 100us (0.1ms) and 1ms is considered to be a typical “long” simulation to discover natural dynamic characteristics. For example, in [3,4], all the systems are simulated for at least 100us, up to 1ms. Thus, we see 100us as a doorsill to become “long”. Since longer simulation is considered to uniformly “better” reflect the real dynamics than the shorter one, we thus omit the shorter cases. Also to illustrate, 100ns MD simulation in the GTT case (in explicit solvent) on a single V100 still takes ~3.87 hrs to finish, let alone 100ns is generally considered “not enough” in standard practice.
>
>
> References:
>
> [1] Arts, Marloes, et al. "Two for one: Diffusion models and force fields for coarse-grained molecular dynamics." Journal of Chemical Theory and Computation 19.18 (2023): 6151-6159.
>
> [2] Gur, Mert, et al. "Conformational transition of SARS-CoV-2 spike glycoprotein between its closed and open states." The Journal of chemical physics 153.7 (2020).
>
> [3] Shaw, David E., et al. "Atomic-level characterization of the structural dynamics of proteins." Science 330.6002 (2010): 341-346.
>
> [4] Lindorff-Larsen, Kresten, et al. "How fast-folding proteins fold." Science 334.6055 (2011): 517-520.
>
>
> We sincerely hope our responses can address your concerns. If you have other questions, we are very glad to discuss them during the rebuttal.

---

> > ### Comment · Reviewer_dLpk · 2023-11-22
> > **Most of the concerns addressed**
> >
> > Thank you for your detailed responses and for answering the concerns.
> >
> > > This observation comes from the structural characteristics of different proteins. (i) Some proteins can show very deterministic structure, an extreme example is the cyclic peptides. In this case, the motions in the protein dynamics are limited, and thus with small ensemble diversity from MD simulations; (ii) On the other hand, other proteins show more dynamic behaviors like distinguishably different states. They can have relatively larger diversity, which can be loyally reflected in the long MD simulations. For example, the SARS-Cov-2 spike protein was found to have transitions between its open and closed states [2]. So it is reasonable to compare with a MD-reference diversity rather than claiming it is “the higher the better”
> >
> > This is a reasonable observation. I believe the manuscript would benefit from clarifying this point when introducing the metric.

---

> > > ### Author Response · Authors · 2023-11-22
> > > **Response to reviewer dLpk**
> > >
> > > Dear reviewer dLpk,
> > >
> > > Thank you for the valuable feedbacks to help us improve the quality and clarity of the manuscript. According to your advice, we have accordingly updated the PDF and incorporated our discussion above into the related chapter (Appendix E - metric: diversity).
> > >
> > > Again, we appreciate your effort for reviewing our paper. If you have any other question, we are always happy to discuss with you.
> > >
> > > Best regards,
> > >
> > > Authors #3973

---

> ### Author Response · Authors · 2023-11-21
> **Look forward to further feedbacks during the reviewer-author discussion period**
>
> Dear Reviewer dLpk,
>
> Thank you for your helpful suggestions and the recognition of the soundness and contribution of our paper. We sincerely appreciate your feedback and have carefully responded to each question point-by-point.
>
> Unlike previous years, there will be no second stage of reviewer-author discussion this year. If you have other question or need any clarification during this period, please feel free to contact us and we are very glad to answer.
>
> Thank you again for your precious time and effort !
>
> Best regards,
>
> Authors #3973

---

### Official Review · Reviewer_BVKt · 2023-11-08

**Soundness:** 3 good
**Presentation:** 2 fair
**Contribution:** 3 good
**Rating:** 6
**Confidence:** 3

**Summary:**

This paper introduces a technique for sampling equilibrium distributions of proteins, eliminating the dependency on costly Molecular Dynamic simulations. The suggested technique utilizes the ESMFold protein embeddings and trains an equivariant denoising diffusion model using samples from the Protein Data Bank, predominantly featuring single folded states (i.e., absolute minima of the equilibrium distribution). During the testing phase, the model diffuses to a specified noise level, partially erasing the protein structure but not entirely, and then it denoises it again, resulting in a different protein conformation. This translation from one structure to another is employed to sample different conformations, starting from the folded structure. The method's effectiveness is assessed using various metrics such as validity (defined as the non-clash) fidelity (defined as the JS between reference and samples TICA distributions) and Diversity. The method seems to outperform previous works.

**Strengths:**

- The motivation and goals of this work are very relevant. Being able to sample protein equilibrium distributions without need of computing expensive molecular dynamics can have a high impact in the sampling community.
- The metrics used in the paper to assess the quality of the equilibrium distributions show the proposed method outperforms previous works.
- The paper contains many metrics assessing different aspects of the generated distributions.

**Weaknesses:**

1)
A more thorough review of prior studies could be beneficial. For instance, EigenFold (Jin et al, 2023) is a diffusion model trained solely on PDB samples with the same aim to generalize to protein distributions. It would be highly useful for readers to have a more detailed comparison between the proposed method and EigenFold, specially considering the apparent superior performance of the proposed method.

1.1 What is novel in the proposed method w.r.t. EigenFold?
1.2 Is the performance gap between EigenFold and the proposed method attributed to a difference in the conceptual approach or is it due to a more technical element such as the use of ESMFold in place of OmegaFold embeddings?

I think answering these questions can really benefit future works when trying to spot the key aspects of the model without need to dig into the codebase.



2)
The validity metrics analyze the non-clash ratio, but it would also be as relevant to examine the distributions of bonds and ensure no bonds are breaking when categorizing a sample as valid. Have the authors conducted this analysis?

**Questions:**

1)
The core part of the proposal of this method is described in 3.2 (forward-backward Dynamcis) where I think a more elaborate explanation could be done here.

For example, when sampling a conformation T ~ p(T | x_0), is x_0 consistently the initiating folded structure, or could it be a T derived from a preceding sampling step?  This is not clear to me from the text. I imagine that if setting it always to the folded structure it would bias the distribution to the minima. Could the authors provide a more precise description of this in the method section?

2)
In section 3.1, could the authors provide more details, or cite relevant literature, explaining how the side chain atoms are derived from the backbone atoms?

3)
In the following sentence "Empirically, increasing T_delta leads to enhanced diversity yet it may hurt exploitation by demanding more reverse steps".

Is this true for any T_delta value? I would suspect that if T_delta goes to a large enough amount of noise (reaching the gaussian distribution), the result would be equivalent sampling from the trained PDB distribution, resulting again in sampling from the folded minima instead of a diverse equilibrium dataset.

4) In section 3.2 (Score Network architecture), the authors indicate that there have been minor modifications to IPA to include pair representations with edge layers. Could the authors provide more context as to why this is necessary and not arbitrary?

5) In Appenix B, Algorithm 2
How is the algorithm returning x_0? Based on the paper wasn't x_0 the starting point?

6) Better interpretation on why the method works:
Given that the model is optimized on the PDB dataset, we would expect it to only learn the PDB landscape. However when degenerating the process with the proposed approach it actually learns to generate samples close to the equilibrium distribution. Because it never had access to a Force Field and only to the folded state it is quite surprising it is able to "make up" that information. Could it be this is only possible because of the ESMFold embeddings? I.e. do these embeddings contain information about the protein landscape beyond the folded structure, and then are you recovering that information present in the ESMFold embeddings with the proposed method? It would be interesting to know the authors interpretation about this.

---

> ### Author Response · Authors · 2023-11-18
> **Response to Reviewer BVKt: Part 1**
>
> We firstly appreciate the insightful questions and suggestions for improving our manuscript. We have added corresponding experiments and discussion in the updated PDF. Our responses to the specific concerns are listed below:
>
> **Weakness:**
>
> **Review and contextualize str2str based on prior studies.**
> - We appreciate this suggestion and have appended a thorough review by contextualizing the prior works, especially the EigenFold in Appendix G.2. To answer the questions:
>    - Similarity: both models leverage PDB for training with a denoising score matching objective. Difference: the EigenFold modeled a mapping from the sequence-encoded latent space (OmegaFold embedding ) to the structure space, while the Str2str delineated solely the structure space and sampled conformations in an amortized manner. During inference, EigenFold performs classic conditional generation while Str2str transforms the unconditional distribution into a translation proposal, from which it samples diverse conformation.
>   - We note that Str2str does not use ESMFold [1] embedding but only the output structure, which is different from EigenFold. We reckon the performance gap is due to the fact that Eigenfold handles a more challenging conditional generative task, where the embedding condition ($s_i, z_{ij}$) is far more complicated than a scalar label (for example in [2]). Without enough training data for each condition, the resulting diversity is thus limited to a large extent. In contrast, Str2str learns the structure landscape from which the conformations are sampled and can perform better.
>
> **Examine bond breaking as validity.**
> - Thanks for raising this great point! We sincerely adopt this suggestion and investigate the bond-breaking ratio for each model. In specific, we have defined and evaluated a new validity metric (”Val-Bond”) showing the ratio of samples w/o violating a statistical upper bound for adjacent Ca-Ca atoms. Relevant contents are updated in Section 4.1, Appendix E & Table [1, S3-7], and the results show that conformations sampled by Str2str are nearly all valid w.r.t. both clash and bonding.
>
> [1] Lin, Zeming, et al. "Evolutionary-scale prediction of atomic-level protein structure with a language model." Science 379.6637 (2023): 1123-1130.
>
> [2] Song, Yang, et al. "Score-Based Generative Modeling through Stochastic Differential Equations." International Conference on Learning Representations. 2020.

---

> ### Author Response · Authors · 2023-11-18
> **Response to Reviewer BVKt: Part 2**
>
> **Questions**
>
> **Clarification of the Forward-backward dynamics.**
> - Firstly, we clarify that the condition is consistently the input across experiments. Here, we only use $p(T|T_0; \theta)$ as a proposal distribution to obtain i.i.d. samples for each test system. A very natural extension echoing this question is that, we can craft a MCMC sampling pipeline by altering the starting point at each MCMC step $i$ and perform sample using $p(x^{[i+1]}|x^{[i]}; \theta)$ where condition $x^{[i]}$ is the sample from the step $i-1$. According to your suggestion, we provide a description of this in the method section (page 4).
>
> **Description of side chain packing.**
> - We are sorry for not making it clear. In section 3.1, the sampling is conceptual and any form of the side-chain modeling can be used. The key idea is to sample atoms step-wise, from backbone to side-chains. In practice, as introduced in Section 3.2,  we used the packing module FASPR [3] which performs a rotamer-library search. Other packing methods can also be used, such as Diffpack [4] that predicts the torsion angles and use them to recover the corresponding atoms similar to what is done in AF2 [5].
>
> **The effect of $T_\delta$ on ensemble diversity.**
> - Thanks for bringing up this question. To investigate this behavior, we additionally investigated the ensemble diversity versus different $T_\delta$ in Appendix F.3 / Figure S7. Indeed, the diversity tends to plateau after $T_\delta$=0.8 for PF and SDE, which shows the enhancement of diversity does not apply to all $T_\delta$. We appreciate this empirical finding and have left a remark accordingly in the paper. Moreoever, from a theoretical view, the forward diffusion will not reach the Gaussian even for T=1.0; it only approximates the Gaussian when $t$ goes up and it is not strictly equivalent to sampling from a gaussian noise.
>
> **The motivation of using edge layers.**
> - We here justify the rationale: the DenoisingIPA can be viewed as a variant of the StructureModule (SM) in AF2 that translates frames to frames. The original SM only updates the single representation $s_i$, frames $T_i$ but not pair representation $z_{ij}$. Since the $z_{ij}$ there is the output of a strong encoder model called Evoformer. We can anticipate very informative $z_{ij}$ even without any updating inside the SM. However in our case, the initial $z_{ij}$ is as simple as the input embedding (time, position). So it is better to update the pair representation along inside $s_i, T_i$ in the model.
>
>
> **Typo in algorithm 2.**
> - Thanks for pointing this out. We have refined the notation in Algorithm 2. Also, we note that a change of variable is applied for the backward time domain $[T_\delta, 2T_\delta]$ so that we can conveniently leverage the prior discretization of time.
>
> **Interpretation on why str2str works.**
> - Thank you for starting us off with this insightful question.
>   - Firstly, we summarize that Str2str learns the vector fields, i.e. score, pointing towards a data-like stuff in the whole structure space (protein landscape). Thus, the trained model can (be expected to) push arbitrary perturbed structure to a nearby data-like minima. Str2str demonstrates a typical transfer learning setting by pretraining on PDB dataset, which enables it to do zero-shot conformation sampling for unseen proteins. Unlike EigenFold, str2str does not rely on sequence embedding but only an initial geometric frame during inference.
>   - “The reason why the model works” can lie in our key assumption that: the dynamic patterns among conformations can be partially shared with the patterns between different structures in the PDB. That is to say, the common inter-protein structure differences in the training set can be distilled in the score network, which is “unlocked” during sampling to infer the inter-conformation deviation within a specific protein system. This agrees with the fundamental motivation of “generalization” in deep learning.
>
> References:
>
> [3] Huang, Xiaoqiang, Robin Pearce, and Yang Zhang. "FASPR: an open-source tool for fast and accurate protein side-chain packing." Bioinformatics 36.12 (2020): 3758-3765.
>
> [4] Zhan, Yangtian, et al. "DiffPack: A Torsional Diffusion Model for Autoregressive Protein Side-Chain Packing." arXiv preprint arXiv:2306.01794 (2023).
>
> [5] Jumper, John, et al. "Highly accurate protein structure prediction with AlphaFold." Nature 596.7873 (2021): 583-589.
>
> We really hope the above responses and revisions can address your concerns. Please kindly let us know if you have any other questions. We’re always happy to have further discussion and improve the quality of the manuscript.

---

> ### Author Response · Authors · 2023-11-21
> **Look forward to further feedback during the reviewer-author discussion period**
>
> Dear Reviewer BVKt,
>
> Thank you for your helpful suggestions and the recognition of the soundness and contribution of our paper. We sincerely appreciate your feedback and have carefully responded to each question point-by-point.
>
> This is a kind reminder that as the reviewer-author discussion period is ending soon, we look forward to hearing from you about your feedback to our response. (Unlike previous years, there will be no second stage of reviewer-author discussion this year)
>
> In particular, we have added more experimental results and discussions per your advice. Please refer to our posted point-to-point response and we are glad to discuss with you.
>
> Thank you again for your precious time and effort !
>
> Best regards,
>
> Authors #3973

---

> > ### Author Response · Authors · 2023-11-22
> >
> > Dear reviewer BVKt,
> >
> > We are grateful for your time for reviewing our paper. We have made clarification on each question and added the experimental results and discussion per your suggestion. Since the deadline of the discussion period is about to end, and other reviewers have given their feedback, we are very expectant to see your re-evaluation of our response above.
> >
> > Thank you very much!
> >
> > Authors #3973

---

### Author Response · Authors · 2023-11-18
**Global response and summary of revisions**

Response to all the reviewers, (senior) area chairs, and readers:

We would like firstly to appreciate the constructive and helpful reviews from all of the reviewers. We’ve carefully considered the suggestion and revised the manuscript accordingly. In specific, we have mainly made the following changes:

- We introduce a new validity metric called Val-bond which evaluates whether adjacent Ca-Ca atoms exceed a certain threshold. The performance of all models and ablation settings are already updated in each table. (Reviewer BVKt weakness 2)
- We append an ablation experiment investigating the effect of rotation loss in Appendix section C.4. (Reviewer Fe1f question 2)
- We extend a discussion section in Appendix section G. which aims to better contextualize the proposed Str2str and compare it with previous methods. (Reviewer BVKt weakness 1; Reviewer 6bVL weakness 2)
- In Appendix section F.3., we complement the experiments for sampling diversity under different $T_\delta$. (Reviewer BVKt question 3)
- In Appendix section F.4, we report the performance on the apo/holo dataset in comparison to EigenFold. (Reviewer Fe1f question 6)
- We fix small typo and potential ambiguity. (Reviewer BVKt question 1& 5; Reviewer 6bVL question 3)

Revisions are colored in dark blue all over the manuscript for better recognition.

---

### Meta-Review · Area_Chair_UaZv · 2023-12-11

**Metareview:**

The paper presents a framework for zero-shot protein conformation sampling as an alternative to Monte Carlo (MC) and Molecular Dynamics (MD) simulations. It introduces a structure-to-structure translation approach, leveraging an amortized denoising score-matching objective trained on general crystal structures without relying on simulation data. Empirical evaluations demonstrate effectiveness in protein conformation sampling while being orders of magnitude faster than traditional MD simulations. The reviewers found that the paper was well-motivated, the proposed method outperformed previous work in many relevant metrics, and it was significant that this performance was archived without relying on simulated data. The reviewers pointed out several weaknesses, including the need for a more thorough review of prior studies and a detailed comparison and description of novelty to similar methods like EigenFold. Also, the benchmark set is somewhat limited, and further testing with a wider range of protein structures could provide a more comprehensive evaluation.

**Justification For Why Not Higher Score:**

The paper presents significant advancements in protein conformation sampling, but the reviewers raised concerns about not having a broader range of benchmark sets to validate its efficacy and novelty.

**Justification For Why Not Lower Score:**

Despite identified areas for improvement, the paper's innovative approach and demonstrated effectiveness in protein conformation sampling justify its acceptance for a poster.

---

### Decision · Program_Chairs · 2024-01-16

Accept (poster)